# Resveratrol intervention attenuates chylomicron secretion via repressing intestinal FXR-induced expression of scavenger receptor SR-B1

Juan Pang [1,2,3,12], Fitore Raka[4,5,12], Alya Abbas Heirali[6,7], Weijuan Shao[2], Dinghui Liu[8], Jianqiu Gu[9], Jia Nuo Feng[2,5,10], Chieko Mineo[11], Philip W. Shaul [11], Xiaoxian Qian[8], Bryan Coburn [6,7], Khosrow Adeli [4,5,7,10] ✉, Wenhua Ling [1] ✉ & Tianru Jin [2,5,7,10] ✉

Two common features of dietary polyphenols have hampered our mechanistic understanding of their beneficial effects for decades: targeting multiple organs and extremely low bioavailability. We show here that resveratrol intervention (REV-I) in high-fat diet (HFD)-challenged male mice inhibits chylomicron secretion, associated with reduced expression of jejunal but not hepatic scavenger receptor class B type 1 (SR-B1). Intestinal mucosa-specific SR-B1$^{-/-}$ mice on HFD-challenge exhibit improved lipid homeostasis but show virtually no further response to REV-I. SR-B1 expression in Caco-2 cells cannot be repressed by pure resveratrol compound while fecal-microbiota transplantation from mice on REV-I suppresses jejunal SR-B1 in recipient mice. REV-I reduces fecal levels of bile acids and activity of fecal bile-salt hydrolase. In Caco-2 cells, chenodeoxycholic acid treatment stimulates both FXR and SR-B1. We conclude that gut microbiome is the primary target of REV-I, and REV-I improves lipid homeostasis at least partially via attenuating FXR-stimulated gut SR-B1 elevation.

Many natural products, available commercially in concentrated formulations, have been shown to possess beneficial metabolic effects and are promoted worldwide for prevention or even treatment of metabolic disorders including type 2 diabetes (T2D), nonalcoholic fatty liver disease (NAFLD), and cardiovascular diseases (CVD). Among them, dietary polyphenols of plant origin including resveratrol, have attracted the most attention of biomedical researchers, drug developers and nutritional scientists. These compounds can improve insulin

[1]Department of Nutrition, School of Public Health, Sun Yat-sen University, Guangzhou, PR China. [2]Division of Advanced Diagnostics, Toronto General Hospital Research Institute, University Health Network, Toronto, ON, Canada. [3]Laboratory of Clinical Pharmacy and Adverse Drug Reaction, West China Hospital, Sichuan University, Chengdu, PR China. [4]Department of Molecular Structure and Function Research Institute, The Hospital for Sick Children, Toronto, ON, Canada. [5]Banting and Best Diabetes Centre, Temerty Faculty of Medicine, University of Toronto, Toronto, ON, Canada. [6]Department of Medicine, Division of Infectious Diseases, University Health Network, Toronto, ON, Canada. [7]Department of Laboratory Medicine and Pathobiology, Temerty Faculty of Medicine, University of Toronto, Toronto, ON, Canada. [8]Department of Cardiology, The Third Affiliated Hospital of Sun Yat-sen University, Guangzhou, PR China. [9]Department of Endocrinology and Metabolism and The Institute of Endocrinology, The First Hospital of China Medical University, Shenyang, PR China. [10]Department of Physiology, Temerty Faculty of Medicine, University of Toronto, Toronto, ON, Canada. [11]Department of Pediatrics, University of Texas Southwestern Medical Center, Dallas, TX, USA. [12]These authors contributed equally: Juan Pang, Fitore Raka. ✉e-mail: k.adeli@utoronto.ca; lingwh@mail.sysu.edu.cn; tianru.jin@utoronto.ca

signaling and energy homeostasis in animal models and human subjects[1–5]. For decades, two fundamental challenges have limited our mechanistic understanding of their functions. Firstly, those polyphenols were often shown to target multiple organs or signaling cascades, without defined receptors. Secondly, the bioavailability of them is extremely low. After consumption of a large amount of a phenolic compound, its maximum plasma concentration rarely exceeds 1 μM. Thus, dietary polyphenols may exert their functions initially in the gut where their concentrations are the highest, possibly by targeting gut microbiota. They may change gut microbiota profiles, regulate production of bacterial products or fermentation products of the food, or exert their functions via gut metabolites of a given polyphenol[6–9]. Indeed, several recent studies have shown that beneficial effects of resveratrol intervention (REV-I) were associated with alterations in gut microbiome[4,10,11].

Intestinal production of ApoB48-containing chylomicrons is significantly increased during insulin resistance, while its clearance is impaired in T2D or NAFLD[12,13]. Chylomicrons are major forms of circulating triglyceride (TG)-rich lipoprotein (TRL), while another one is very low-density lipoprotein (VLDL), produced in the liver. The major lipid composition of chylomicrons is TG, followed by dietary cholesterol in the form of cholesteryl ester. Studies have shown that postprandial hypertriglyceridemia is a powerful predictor of CVD[14]. Chylomicron remnants after lipolysis can enter the arterial intima, and remnant cholesterol can accumulate in intimal foam cells to form atherosclerotic plaques[15,16]. Chylomicron remnants can also be taken up by the liver, raising the risk of NAFLD[17,18].

Chylomicron production occurs in enterocytes and involves dietary fatty acid (FA) uptake, TG synthesis, chylomicron assembly, and secretion across the basolateral membrane[19,20]. Multiple lipid transporters participate in the process, including cluster determinant 36 (CD36)[21], scavenger receptor class B type 1 (SR-B1)[22], and FA transport protein 4 (FATP4)[23]. Among them, SR-B1 was shown to be elevated the most in the small intestine of insulin-resistant hamster model, associated with increased postprandial TG and TRL accumulation[22]. SR-B1 knockout (KO) mice or mice treated with SR-B1 inhibitor displayed less chylomicron production[22,24].

Here we report a previously unrecognized function of REV-I: inhibition of gut chylomicron secretion. Gut but not liver SR-B1 was shown to be downregulated by REV-I. Importantly, SR-B1 repression could not be observed by resveratrol treatment in the Caco-2 cells. We then observed that Caco-2 cells treated with fecal extract (FE) from mice on HFD plus REV-I showed relatively lower levels of SR-B1, when compared with Caco-2 cells treated with FE from HFD-fed mice without REV-I. Fecal microbiota transplantation (FMT) revealed the effect of feces from mice that received REV-I on improving fat tolerance and reducing jejunal SR-B1. Finally, we conducted metabolomics analyses of FE and gut microbiome analyses, leading to the recognition that bile acid/FXR activation can stimulate jejunal SR-B1 expression while REV-I attenuates HFD challenge-induced fecal bile acid elevation.

## Results

### REV-I reduces chylomicron production in HFD-challenged mice
As shown (Fig. 1a), 6-week-old C57BL/6J male mice were fed with LFD, HFD, or HFD + REV-I (HFR) for 8 weeks. REV-I reduced HFD-induced body weight gain (Fig. 1b, c) and decreased fat mass and hepatic fat content, but not liver/body weight ratio (Supplementary Fig. 1a–c). There was no difference in food intake between HFD and HFR groups (Supplementary Fig. 1d). REV-I improved glucose and insulin tolerance, accompanied by reduced fasting glucose and insulin levels (Fig. 1d–g). When adjusted for basal blood glucose level, the improvement in insulin tolerance was virtually absent in our current experimental settings (Supplementary Fig. 1e). Furthermore, REV-I reduced the levels of fasting TG but not non-esterified FA (NEFA), which is mainly derived from adipocytes via FA lipolysis (Fig. 1h, i).

Plasma TG level is determined by the balance among chylomicron production, hepatic VLDL production and TRL clearance. When lipoprotein lipase (LPL) activity was blocked with poloxamer 407 in overnight-fasted mice, there was no difference in plasma TG levels among the three groups of mice (Supplementary Fig. 1f), indicating that REV-I in current experimental settings did not significantly affect hepatic VLDL production during fasting. Consistently, REV-I did not attenuate HFD-induced fasting ApoB48 and TG content in TRL (Supplementary Fig. 1g, h). Moreover, plasma LPL activity was similar among the three groups of mice (Supplementary Fig. 1i). In the postprandial condition, plasma TG accumulation in mice challenged with olive oil was reduced by REV-I (Fig. 1j). Although AUC in HFR mice appeared to be even lower than that in LFD mice, the difference did not reach statistical significance. Chylomicron production, determined by assessing TG content and ApoB48 level in isolated TRL, was reduced in mice with REV-I (Fig. 1k, l). These observations suggest that REV-I reduces plasma TG by inhibiting chylomicron production in the fed state. Figure 1m summarizes metabolic effects of REV-I observed in this set of mice.

### REV-I reduces jejunal but not liver SR-B1
Dietary lipid intake and de novo lipogenesis result in gut lipid droplet accumulation, which is then packaged as chylomicrons, or goes through FA β-oxidation. Chylomicrons are secreted through the basolateral membrane into the lacteals, where they join lymph to become chyle (Fig. 2a). In a separate set of mice with or without 8-week REV-I, we measured TG intake and its fecal output. HFD challenge led to elevated TG intake and fecal TG output, but REV-I did not reduce TG intake or increase its output (Fig. 2b, c), suggesting that REV-I did not significantly affect intestinal TG absorption. TG accumulation in the intestine remained at comparable levels (Fig. 2d). We then measured jejunal expression of genes that are associated with lipid metabolism. REV-I did not affect expression of genes that are related to TG uptake, de novo lipogenesis, or chylomicron assembly, but attenuated the stimulation of HFD on expression of Scarb1 (which encodes SR-B1) and increased expression of genes that are responsible for lipolysis and FA β-oxidation (Fig. 2e and Supplementary Table 4). Importantly, jejunal but not liver SR-B1 protein level was reduced by 54% by REV-I (Fig. 2f). SR-B1 at gut apical membrane mediates lipid sensing and chylomicron secretion[25]. We show here that REV-I also attenuated apical migration of SR-B1 (Fig. 2g, h).

### Intestinal mucosa-specific SR-B1 KO mice show lack of further response to REV-I
To further explore the involvement of gut SR-B1, experiments were then conducted in intestinal mucosa-specific SR-B1 KO (iScarb1⁻/⁻) mice (Fig. 3a). iScarb1⁻/⁻ and control Scarb1fl/fl mice were fed with HFD, without or with REV-I for 8 weeks. Supplementary Fig. 2a–d shows our validation on intestinal mucosa-specific KO of SR-B1 and observations that at the age of 6 weeks, basal body weight and fasting TG level were comparable between iScarb1⁻/⁻ and control mice. Following HFD challenge, iScarb1⁻/⁻ mice showed lower body weight gain, compared to Scarb1fl/fl mice (Fig. 3b, c). Concomitant REV-I for 8 weeks reduced body weight gain in control but not iScarb1⁻/⁻ mice (Fig. 3c). In control but not iScarb1⁻/⁻ mice, REV-I reduced white fat mass, while liver weight to body weight ratio of the four groups of mice was comparable (Supplementary Fig. 2e, f). Figure 3d–f shows that REV-I resulted in improved glucose and insulin tolerance in iScarb1fl/fl mice but no further improvements by REV-I were observed in iScarb1⁻/⁻ mice. Importantly, iScarb1⁻/⁻ mice showed lower fasting and postprandial plasma TG accumulation (Fig. 3g, h) as well as lower TG and ApoB48 levels in TRL (Fig. 3i, j). However, REV-I generated no further reduction in these parameters in iScarb1⁻/⁻ mice (Fig. 3g–j). Moreover, REV-I enhanced expression of Cpt1a and Acadm in Scarb1fl/fl mice but not in iScarb1⁻/⁻ mice (Fig. 3k). Figure 3l summarizes results of REV-I in these

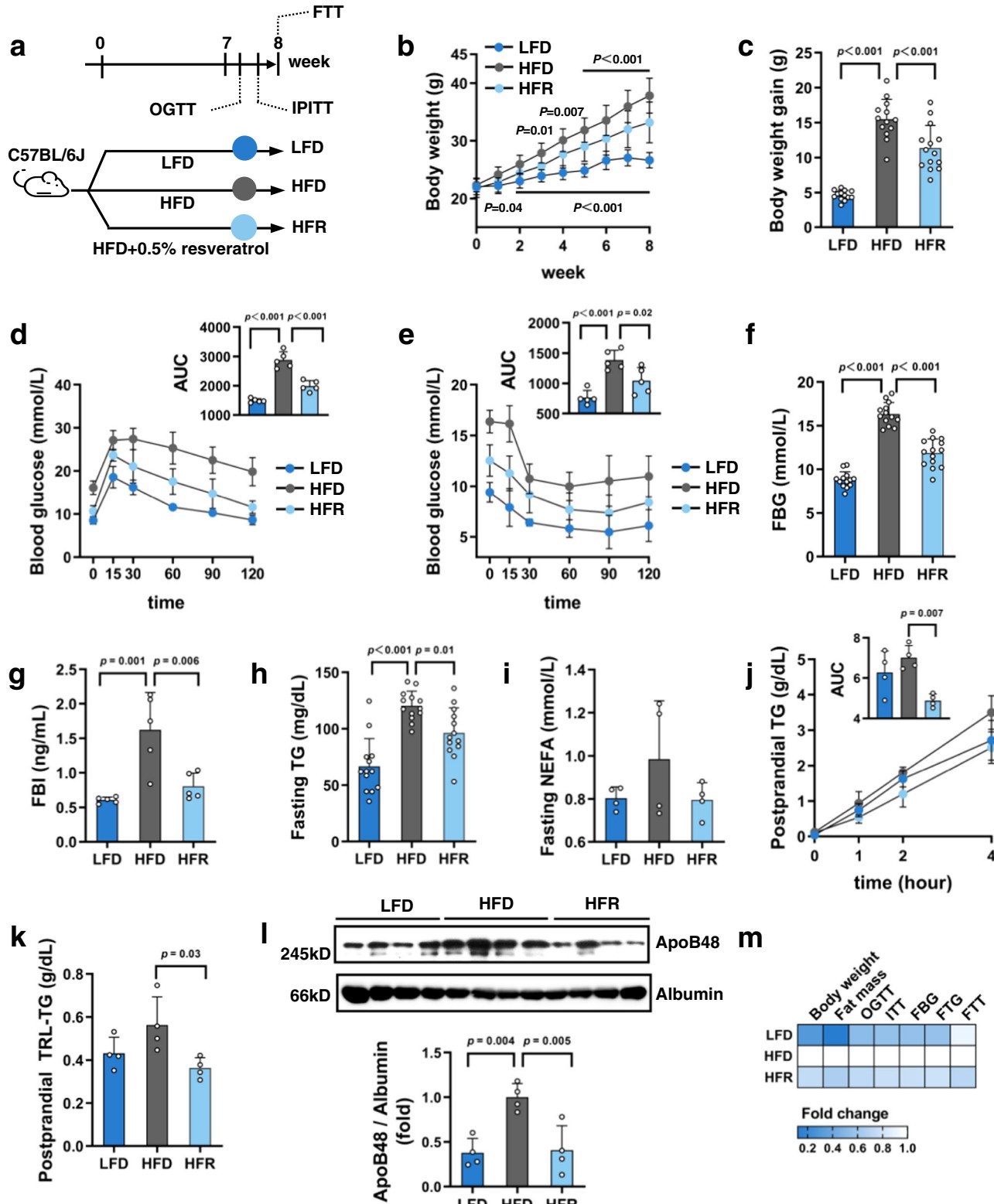

**Fig. 1 | REV-I for 8 weeks reduces chylomicron production in HFD-challenged mice. a** Diagram shows experimental procedures. Body weight change (**b**) and body weight gain (**c**) of mice after 8-week treatment; $n = 13$ for the LFD and HFD groups and $n = 14$ for the HFR group. Blood glucose level and area under the curve (AUC) during OGTT (**d**) and IPITT (**e**); $n = 5$. Fasting blood glucose (FBG) (**f**) ($n = 13$–$14$ as **c**), fasting blood insulin (FBI) (**g**) ($n = 5$), TG (**h**) ($n = 13$–$14$ as **c**), and non-esterified FA (NEFA) levels (**i**) ($n = 4$). **j** Postprandial TG levels during FTT; $n = 4$. **k**, **l** Postprandial plasma collected 4 h after olive oil gavage was ultracentrifuged for isolating TRL (mainly chylomicron). TG concentrations were then measured (**k**), while ApoB48 levels were assessed by Western blotting (**l**); $n = 4$. **m** Heatmap summarizes metabolic effects of REV-I presented in this figure (fold change of a given parameter vs. that in HFD-fed mice which is defined as 1-fold). Statistical significance was evaluated by two-sided one-way ANOVA with Dunnett's post hoc test (compared with HFD group). See also Supplementary Fig. 1. Data are presented as mean ± SD. Source data are provided as a Source Data file.

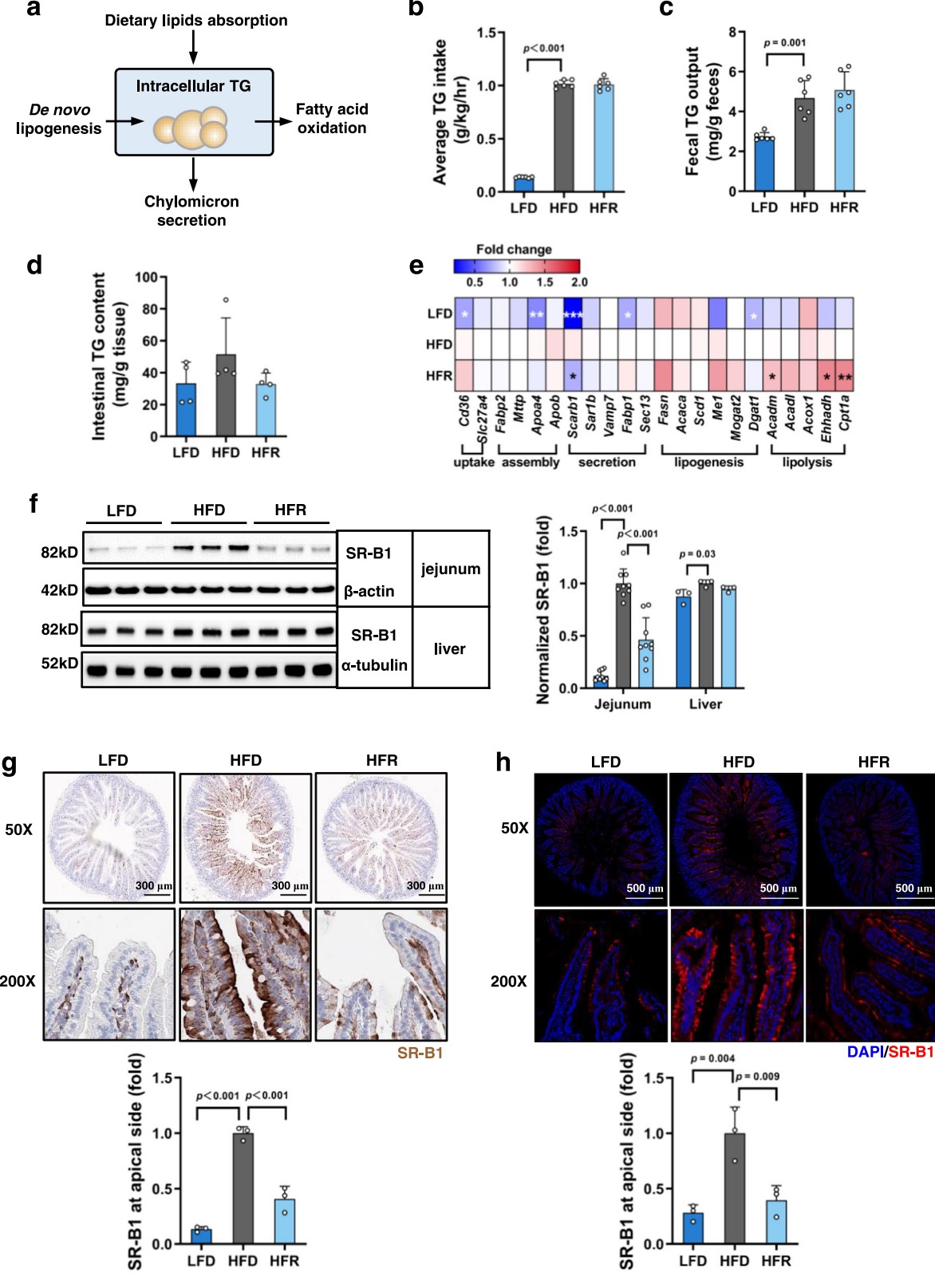

**Fig. 2 | REV-I reduces jejunal but not liver SR-B1. a** Diagram summarizes metabolic processes involved in intestinal lipid (TG) homeostasis. Average TG intake (**b**) and daily fecal TG loss (**c**) of mice individually housed in metabolic cages at the end of the 8th week; $n = 6$. **d** Jejunal intracellular TG contents; $n = 4$. **e** Heatmap shows the expression of genes involved in intestinal lipid metabolism among the three groups of mice (see statistical analysis results of qRT-PCR in Supplementary Table 4); $n = 12$. **f** Jejunal ($n = 9$) and hepatic ($n = 3$) SR-B1 levels were assessed by Western blotting. **g** Representative jejunum SR-B1 immunostaining images (brown), along with its quantitative scores at the apical membrane. The scale bar is 300 μm; $n = 3$. **h** Representative jejunum SR-B1 immunofluorescence staining images (red), along with its quantitative scores at the apical membrane. The scale bar is 500 μm; $n = 3$. Statistical significance was evaluated by two-sided one-way ANOVA with Dunnett's post hoc test (compared with HFD group). *$p < 0.05$, **$p < 0.01$, ***$p < 0.001$. Data are presented as mean ± SD. Source data are provided as a Source Data file.

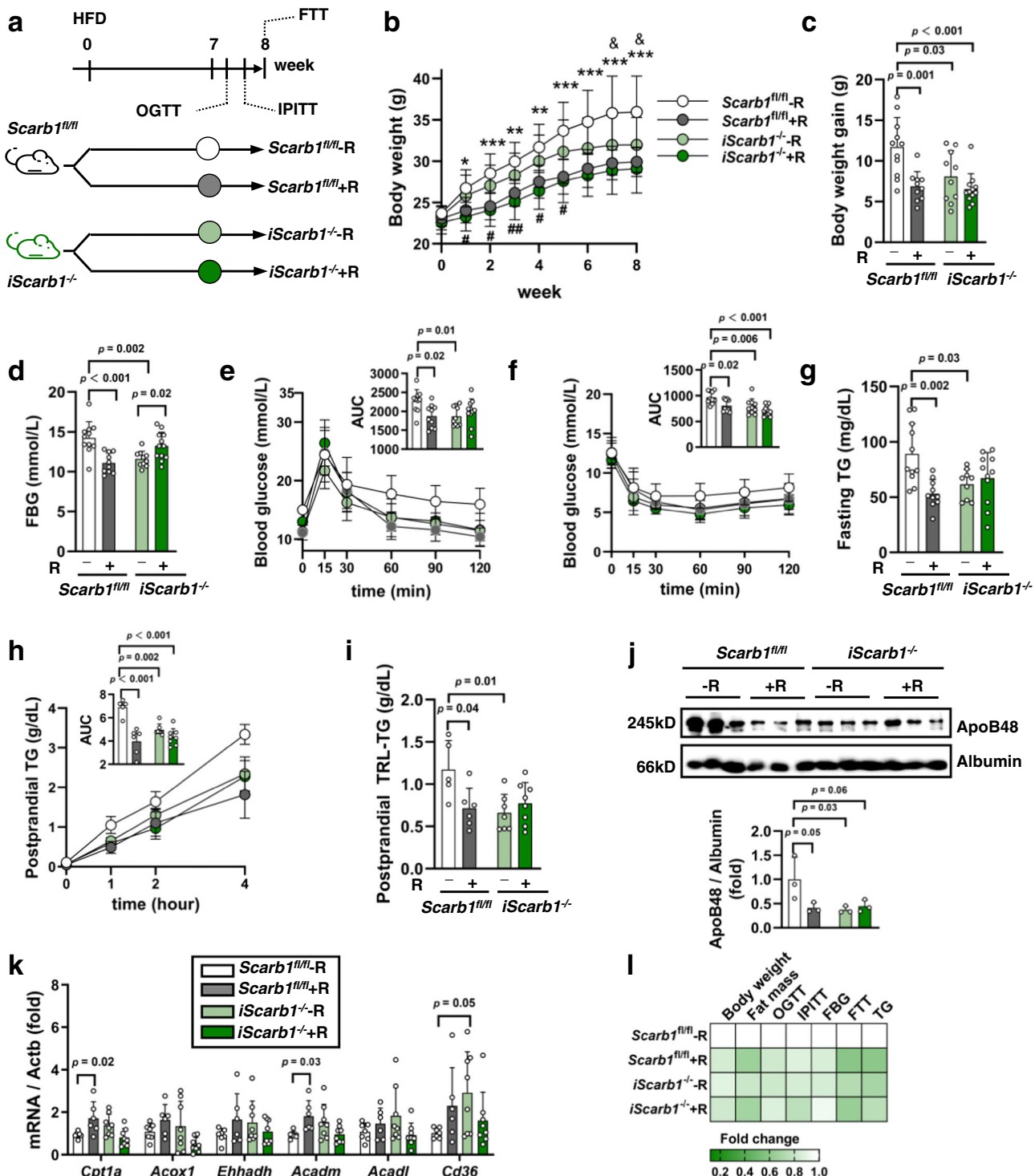

**Fig. 3 | Intestinal mucosa-specific SR-B1 KO mice show lack of further response to REV-I. a** Diagram shows experimental procedures in *Scarb1^fl/fl* and *iScarb1^−/−* mice. Body weight (**b**) and body weight gain (**c**) at the end of the 8th week; *n* = 10 for the *iScarb1^−/−* − R and *Scarb1^fl/fl* + R groups, *n* = 11 for the *Scarb1^fl/fl* − R group and *n* = 12 for the *iScarb1^−/−* + R group. **d** Fasting blood glucose (FBG) levels at the end of the 8th week; *n* = 10–12 as (**b**, **c**). Blood glucose levels and AUC during OGTT (**e**) and IPITT (**f**), *n* = 10–12 as (**b**, **c**). **g** Fasting TG levels at the 8th week; *n* = 10–12 as (**c**). **h** Postprandial TG levels during FTT; *n* = 5 for the *Scarb1^fl/fl* − R group, *n* = 6 for the *Scarb1^fl/fl* + R group, *n* = 7 for the *iScarb1^−/−* − R group and *n* = 8 for the *iScarb1^−/−* + R group. **i, j** Postprandial plasma collected 4 h after olive oil gavage was

ultracentrifuged for isolating TRL. TG concentrations were then measured (**i**), and ApoB48 levels were assessed (**j**) (albumin as loading control); *n* = 3. **k** Expression of *Cd36* and other genes that are involved in FA β-oxidation; *n* = 5–8 as (**h**). **l** Heatmap summarizes metabolic effects of REV-I presented in this figure. A given parameter in *Scarb1^fl/fl* control mice without REV-I is defined as one-fold. Statistical significance was evaluated by two-sided two-way ANOVA with Šidák post hoc test. For (**b**), *, **, or ***, *Scarb1^fl/fl* + R vs. *Scarb1^fl/fl* − R; # or ##, *iScarb1^−/−* + R vs. *iScarb1^−/−* − R; &, *Scarb1^fl/fl* − R vs. *iScarb1^−/−* − R. See also Supplementary Fig. 2. Data are presented as mean ± SD. Source data are provided as a Source Data file.

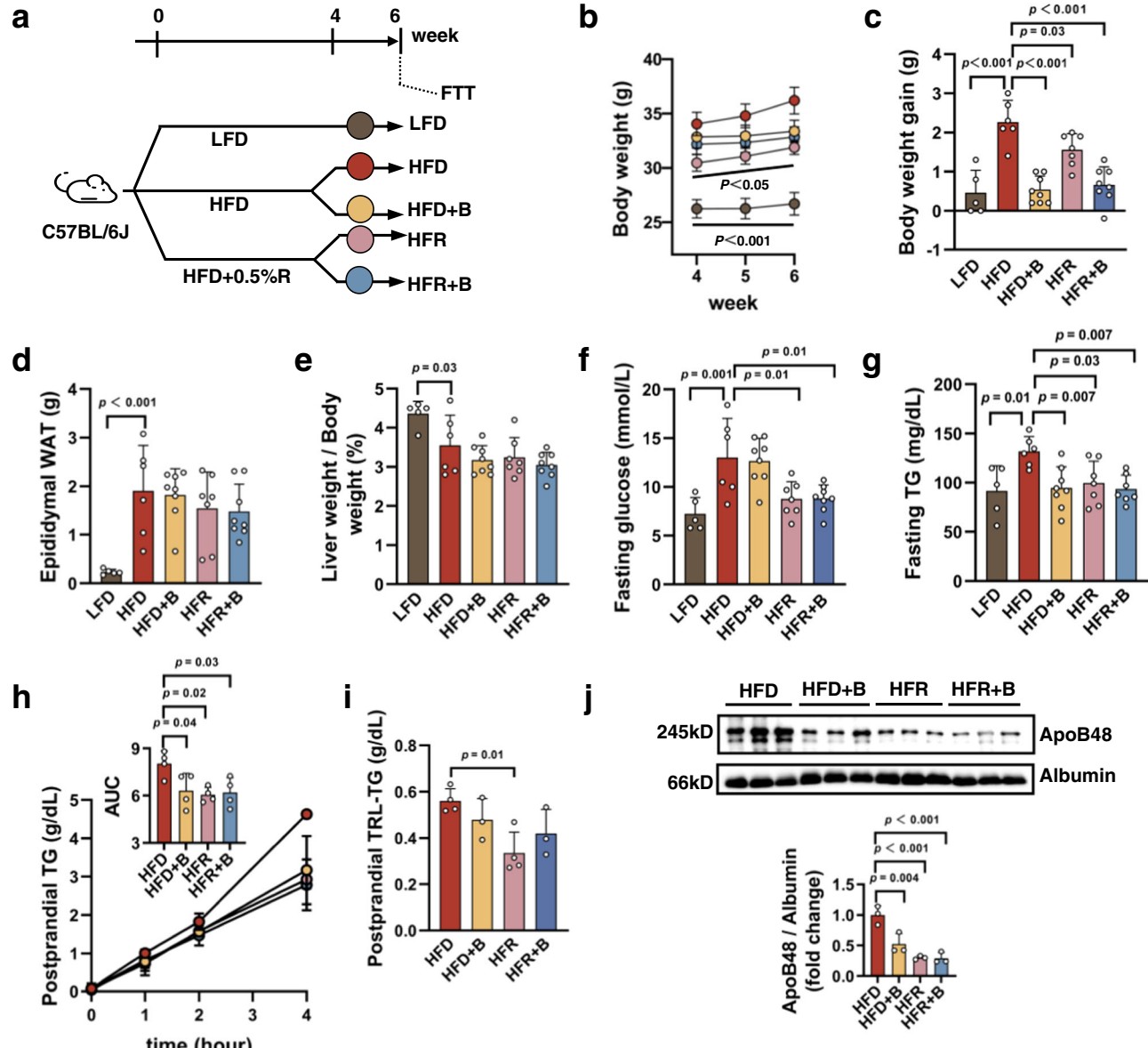

**Fig. 4 | BLT-1 gavage generates no additive effect with REV-I. a** Diagram shows the experimental procedures in C57BL/6J mice treated with resveratrol (HFR), or BLT-1 (B), or both (HFR + B). Body weight (**b**) and body weight gain (**c**) from week 4–6; $n = 5$ for the LFD group, $n = 6$ for the HFD group, $n = 7$ for the HFR group and $n = 8$ for the HFD + B and HFR + B groups. Epididymal white adipose tissue (**d**), liver weight to body weight ratio (**e**), fasting glucose (**f**), and TG levels (**g**) at the end of the 6th week; $n = 5$–8 as (**b**, **c**). **h** Plasma TG levels during FTT; $n = 4$. **i**, **j** Postprandial plasma collected 4 h after olive oil gavage was ultracentrifuged for isolating TRL. TG concentrations were then measured (**i**), and ApoB48 levels were assessed by Western blotting (**j**); $n = 3$. Statistical significance was evaluated by two-sided one-way ANOVA with Dunnett's post hoc test (compared with the HFD group). *$p < 0.05$, **$p < 0.01$, ***or ###$p < 0.001$. For (**b**), *, HFD vs. HFR; ###, LFD vs. HFD. Data are presented as mean ± SD. Source data are provided as a Source Data file.

four groups of mice, suggesting that gut SR-B1 is a major target of REV-I in improving energy homeostasis and reducing chylomicron secretion.

## BLT-1 gavage generates no additive effect with REV-I

We then further tested the role of gut SR-B1 in mediating effects of REV-I by orally applying the SR-B1 inhibitor BLT-1. C57BL/6J mice were fed with LFD, HFD, or HFR for 4 weeks (Fig. 4a). The HFD and HFR groups were then further divided into two subgroups, without or with daily BLT-1 gavage for 2 more weeks. BLT-1 or REV-I attenuated HFD-induced body weight gain, but they generated no additive effect (Fig. 4b, c). Two-week BLT-1 gavage, or 6-week REV-I, or combined use of them, did not prevent epididymal fat mass increase induced by HFD, and did not change liver weight to body weight ratio (Fig. 4d, e). Furthermore, BLT-1 did not reduce fasting-glucose level, while REV-I

did (Fig. 4f). Mice in all treatment groups showed decreased fasting and postprandial TG levels and reduced chylomicron production (Fig. 4g–j). Similarly, the combined use of REV-I and BLT-1 generated no additive effect (Fig. 4g–j).

## REV-I represses jejunal SR-B1 involving reduced transcriptional activity of NF-κB-p65

To explore how REV-I can downregulate SR-B1, we located transcriptional factor binding motifs on *Scarb1* promoter, including that for SREBP-1. Western blotting, however, showed comparable jejunal SREBP-1 levels in LFD, HFD, and HFR groups (Fig. 5a). Although qRT-PCR showed elevated jejunal *Srebf1* in HFD mice, REV-I did not repress it and generated no effect on expression of SREBP-1 downstream targets (Fig. 5b). Within human and rodent *SCARB1* or *Scarb1* promoters,

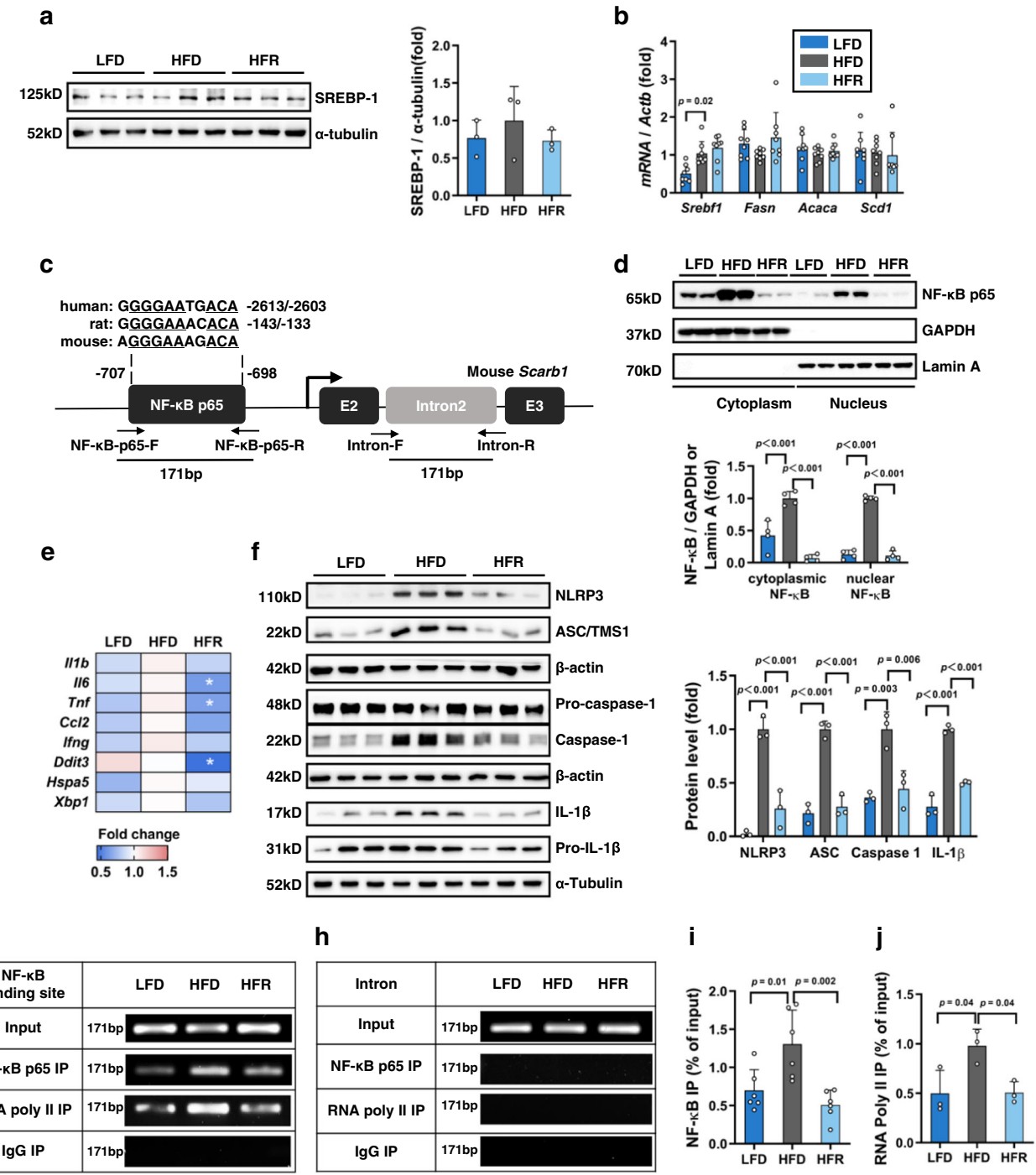

**Fig. 5 | REV-I represses jejunal SR-B1 involving reduced transcriptional activity of NF-κB-p65. a** Jejunum expression of SREBP-1 detected by Western blotting; $n = 3$. **b** Comparison of expression of *Srebf1* (which encodes SREBP-1) and its target genes in jejunum; $n = 8$. **c** Location of the conserved NF-κB p65 binding motif within human, rat and mouse *SCARB1/Scarb1* promoters and nucleotide primers utilized in ChIP and qChIP. **d** Detection of cytoplasmic and nucleus NF-κB-p65 by Western blotting in the jejunum of indicated groups of mice, presented in Fig. 1; $n = 4$. The blot of NF-κB-p65 was stripped to re-probe for Lamin A. **e** Heatmap shows the comparison of intestinal inflammatory genes that are known to be regulated by NF-κB in the jejunum; $n = 8$ (see statistical analysis results of qRT-PCR in

Supplementary Table 5). **f** Western blotting shows the relative expression of inflammasome components in the jejunum. $n = 3$. The blot of Pro-caspase-1 was stripped and re-probed for β-actin. ChIP shows the binding of NF-κB or RNA Polymerase II to the mouse *Scarb1* promoter (**g**) ($n = 6$) but not the intron region (**h**) ($n = 3$) in the jejunum. qChIP shows the comparison of binding of NF-κB (**i**) ($n = 6$) or RNA Polymerase II (**j**) ($n = 3$) to mouse *Scarb1* promoter in the jejunum. Statistical significance was evaluated by two-sided one-way ANOVA with Dunnett's post hoc test (compared with the HFD group). *$p < 0.05$, **$p < 0.01$, ***$p < 0.001$. Data are presented as mean ± SD. Source data are provided as a Source Data file.

we located a conserved binding motif for NF-κB-p65 (Fig. 5c). Eight-week HFD challenge increased jejunal NF-κB-p65 expression, while REV-I blocked the increase (Fig. 5d). Such blockage was associated with inhibition of genes that are members of the NF-κB signaling (Fig. 5e and

Supplementary Table 5) and key members of NLRP3 inflammasome (Fig. 5f). With the ChIP approach, we observed increased interactions between NF-κB-p65 or RNA Polymerase II with the *Scarb1* promoter in mice on HFD, while REV-I attenuated their interactions (Fig. 5g–j).

## Inhibition of REV-I on gut SR-B1 involves gut microbiome

In Caco-2 cells treated with resveratrol at dosages not affecting cell viability (1, 5, and 25 μM), we observed no repression of SR-B1 or NF-κB-p65 at protein or mRNA levels (Supplementary Fig. 3a–c). We then established the intestinal barrier model by culturing Caco-2 in microporous PET transwell inserts for 21 days, allowing cells to be differentiated into monolayers with polarity (Supplementary Fig. 3d). Lipid micelle was shown to stimulate chylomicron secretion, detected by measuring BODIPY labeled $C_{12}$ FA secreted into the lower compartment. Resveratrol treatment, however, did not reduce its elevation (Supplementary Fig. 3e). The discrepancy between in vivo and in vitro effects of resveratrol on SR-B1 suggests that jejunal SR-B1 regulation involves entities that are absent in vitro, such as gut microbiota. We hence collected fresh feces from mice fed with HFD or HFR for 8 weeks. Following sonication and filtration, we collected sterile FEs. When HFD-FE or HFR-FE was diluted 150-fold or higher, we did not see their repression on Caco-2 viability (Supplementary Fig. 4a, b). When those FEs (diluted at 1:1200) were applied to Caco-2-derived intestinal barrier model, HFR-FE treated cells showed relatively lower levels of chylomicron secretion (Supplementary Fig. 4c). Furthermore, the stimulatory effect of HFD-FE on SR-B1 and NF-kB p65 was absent when HFR-FE was applied (Supplementary Fig. 4d, e). We then treated sterile FEs at 95 °C for 5 min before they were applied to Caco-2. As shown (Supplementary Fig. 4f), heated HFR-FE treated Caco-2 cells showed lower levels of SR-B1 when compared with Caco-2 treated with heated HFD-FE. Hence, regulatory effect of REV-I on gut NF-κB and SR-B1 likely involves certain heat-stable products in feces.

Thereafter, we conducted a 12-day FMT, in which mice received FMT in the form of a fecal slurry from HFD-fed mice (HFD-FMT), HFR-treated mice (HFR-FMT), or HFD-fed mice plus resveratrol gavage (500 mg/kg body weight) on day 1, 3 and 5 (HFD-FMT + R) (Fig. 6a). We designed this short-term FMT, aimed to ask whether repression on gut SR-B1 can occur ahead of the known effects of FMT from REV-I treated mice on lowering body weight and glucose homeostasis[4]. As shown, on day 7, we did not see altered glucose tolerance within either LFD or HFD group (Fig. 6b) and FMT did not affect body weight gain among mice in LFD or HFD groups (Fig. 6c, d). We also did not see an effect of given FMT on fasting TG levels yet (Fig. 6e). When FTT was conducted on day 12, in which mice were challenged with olive oil after the blockage of plasma LPL activity, in the LFD groups, mice received HFD-FMT + R showed the highest levels of postprandial TG (Fig. 6f). Although we cannot provide a conclusive explanation for such elevation yet, this observation suggests that lipid homeostatic effect of resveratrol cannot be achieved in a short-term on its own (3-day gavage), because longer time is required for resveratrol to "re-shape" gut microbiome. Importantly, in HFD groups, mice that received HFR-FMT showed lower postprandial TG levels when compared with mice received HFD-FMT (Fig. 6f). In isolated postprandial TRL, HFD-fed mice that received HFR-FMT had lower TG and ApoB48 levels when compared with those received HFD-FMT, although the difference did not reach statistical significance (Fig. 6g, h). Finally, jejunal SR-B1 levels in HFD-fed mice receiving HFR-FMT were lower than that received HFD-FMT (Fig. 6i).

## REV-I attenuates HFD-induced fecal bile acid elevation

We moved to explore changes in feces that are associated with the repression of jejunal SR-B1 following REV-I. Firstly, we asked whether two commercially available gut microbial metabolites of resveratrol, DHR and lunularin (Supplementary Fig. 5a), repress SR-B1. In Caco-2 cells pre-treated with LPS and IFN-γ, DHR exerted anti-inflammatory effect, as it inhibited expression of *IL6* and *TNF* (encodes human IL-6 and TNF-α, respectively). The treatment, however, did not reduce expression of *SCARB1* and *RELA* in the absence or presence of LPS and IFN-γ pre-treatment (Supplementary Fig. 5b–d).

We then performed untargeted metabolomics profiling of FEs from HFD and HFR mice, without including LFD group as we were mainly planning to explore the involvement of heat-stable products. Hence, HFR-FE samples were treated without or with 95 °C heating for 5 min, and the latter was referring as HRH-FE. Through LC-MS/MS analysis, we detected 2053 metabolites (Supplementary Data 1). Principal component analysis (PCA) revealed that composition of metabolites in HFD-FE and HFR-FE are different, and after heat treatment, a large portion of metabolites in HFR-FE was transformed (Supplementary Fig. 6a). We then conducted differential analysis and identified 238 differential metabolites between HFR and HFD groups (Supplementary Data 2), and 432 differential metabolites between HRH and HFD groups (Supplementary Data 3). Supplementary Data 4 lists 30 key differential metabolites with their retention times. The top 10 increased and decreased metabolites in the HFR group, compared with the HFD group, are shown in Supplementary Fig. 6b. Biological activities of those metabolites remained largely unknown. We picked 4'-hydroxyflavone for a test in Caco-2 cells as this metabolite was reported to inhibit SREBP-1[26]. As shown, although 4'-hydroxyflavone repressed expression of genes that encode IL-6, IL-1β and MCP-1, it did not repress *SCARB1* or *RELA* (Supplementary Fig. 6c).

Kyoto Encyclopedia of Genes and Genomes (KEGG) pathway enrichment analysis was performed. Our results suggest that sphingolipid metabolism and bile acid metabolism-related pathways were mostly modulated by REV-I (Supplementary Fig. 6d). As FE from HFR-fed mice before or after heating showed a comparable effect on SR-B1 expression, we looked for compounds with comparable levels in HFR-FE and HRH-FE. K-Means clustering analysis was performed in which differential compounds were divided into seven subclasses according to their relative abundance in different groups. Only subclass 3 showed comparable reduced levels in HFR-FE and HRH-FE when they were compared with that in the HFD group (Supplementary Fig. 6e). Subclass 3 mainly contains sphingolipids, bile acids (BA) and certain amino acids. Through differential analysis, levels of chenodeoxycholic acid (CDCA) and deoxycholic acid (DCA) were shown to be reduced in both HFR-FE and HFH-FE when compared with that in HFD-FE (Fig. 7a, b). The levels of lithocholic acid (LCA) were also lower in HFR-FE and HRH-FE than that in HFD-FE, but the difference did not reach statistical significance level (Supplementary Data 2 and 3).

We then measured levels of certain BAs in mouse feces and serum and included samples from LFD-fed mice. As shown, HFD feeding increased fecal total BA, CDCA and DCA levels, while these increases were prevented by REV-I (Fig. 7c, d). Apparently, heating generated no effect on their measurement. In serum, although HFD or REV-I did not alter total BA levels (Fig. 7e), HFD-feeding increased unconjugated BAs including cholic acid (CA), CDCA and LCA, and decreased conjugated BAs including TCA, TαMCA and TDCA. Importantly, effects of HFD on CDCA and TCDCA were reversed by REV-I (Fig. 7e, f).

BAs are synthesized in the liver, conjugated to taurine or glycine before exporting to the intestine, where BSH produced by gut bacteria deconjugates BAs into unconjugated ones[27]. The mRNA levels of hepatic BA synthetic genes, including *Cyp7a1*, *Cyp7b1* and *Cyp8b1*, were reduced by HFD, while REV-I restored their expression (Fig. 7g). Importantly, expression of *Cyp2c70*, which encodes the key enzyme for converting CDCA into muricholic acid (MCA)[28], was repressed by HFD and elevated by REV-I (Fig. 7g). Consistently, hepatic CDCA level was decreased following REV-I (Fig. 7h). We have also measured BSH activity in mouse feces and observed that REV-I attenuated BSH activity which was elevated by HFD challenge (Fig. 7i).

We then compared gut microbiome composition in mice on HFD, without or with REV-I. Supplementary Fig. 7a shows the difference of Bray–Curtis beta-diversity in the two groups visualized by three-dimensional principal coordinate analysis (PCoA). At the phylum level, there was a decreasing tendency in the relative abundance of *Firmicutes* while the relative abundance of *Bacteroidetes* and the

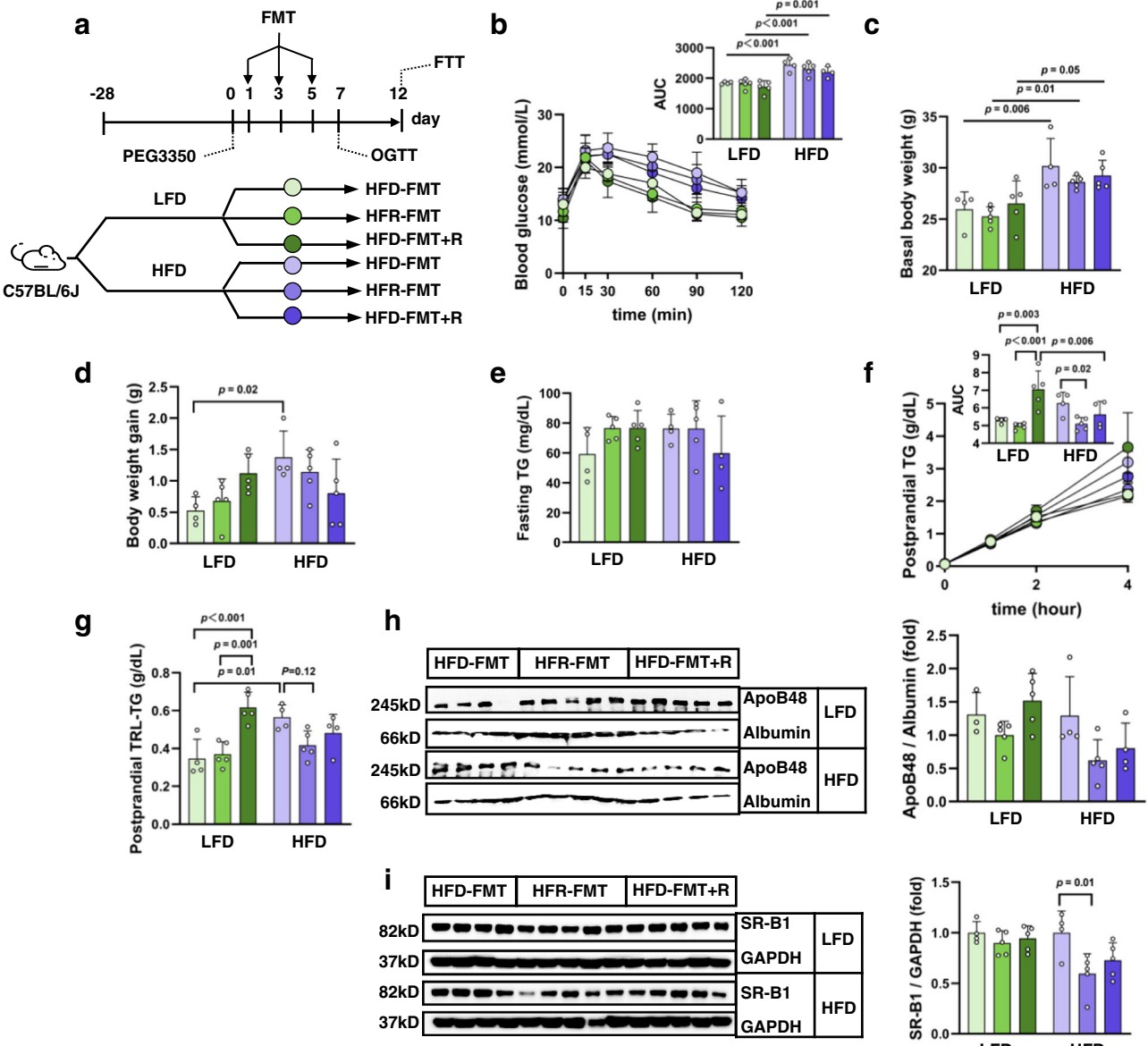

**Fig. 6 | The inhibitory effect of REV-I on SR-B1 involves gut microbiota.**
**a** Diagram shows the design of the short-term FMT. Six-week-old male mice on LFD or HFD for 4 weeks were further divided randomly into three groups, receiving indicated FMT following PEG3350 treatment. **b** Blood glucose level and AUC during OGTT on day 7. **c** Basal body weight before FMT. **d** Body weight gain between day 0 and day 12. **e** Fasting TG level on day 12. **f** Postprandial TG level during FTT on day 12. **g**, **h** Postprandial plasma collected 4 h after olive oil gavage was ultracentrifuged to isolate TRL. TG concentrations were then measured (**g**), and ApoB48 levels were assessed by Western blotting (**h**). **i** Jejunal SR-B1 level. $n = 4$ for the LFD-HFD-FMT and HFD-HFD-FMT groups and $n = 5$ for the other groups in the above test. Statistical significance was evaluated by two-sided two-way ANOVA with Šidák post hoc test. *$p < 0.05$, **$p < 0.01$, ***$p < 0.001$. See also Supplementary Figs. 3 and 4. Data are presented as mean ± SD. Source data are provided as a Source Data file.

ratio of *Bacteroidetes/Firmicutes* were increased in the HFR group (Supplementary Fig. 7b, c). Interestingly, at the genus level, BSH-enriched bacteria such as *Lactobacillus*, *Bifidobacterium*, *Clostridium* and *Enterococcus* were all repressed by REV-I (Supplementary Fig. 7d). We also profiled gut microbiome with mouse cecum contents. Cecum content samples were separated based on diet in terms of Bray–Curtis beta-diversity (Supplementary Fig. 7e). Shannon Diversity index, inverse Simpson and observed taxa metrics were utilized to analyze alpha-diversity differences. Cecum samples from HFD-fed mice showed much lower alpha-diversity when compared with LFD or HFR mice (Supplementary Fig. 7f). Supplementary Fig. 7g shows the taxonomic summary of phyla in the three groups of mice. In contrast to microbiome profiling of fecal samples, we did not see elevated *Bacteroidetes/Firmicutes* ratio with REV-I

(Supplementary Fig. 7g). Consistently, BSH-producing bacteria including *Bacteroides*, *Clostridium* and *Enterococcus* were reduced in the HFR group (Supplementary Fig. 7h). Thus, in addition to repressing liver BA production, REV-I may reduce fecal BA level by inhibiting BSH-producing gut microbiome.

## CDCA or GW4064 upregulates SR-B1 in Caco-2 cells while GW4064 blocks the in vivo function of REV-I on SR-B1

We picked CDCA for further study in Caco-2 cells, as it is the most potent natural agonist of Farnesoid X Receptor (FXR). A previous study showed that FXR activation upregulated hepatic SR-B1 expression[29]. We ask whether FXR activates SR-B1 in Caco-2 cells. Both CDCA and GW4064, a synthetic FXR agonist, were shown to activate *NR1H4* (which encodes human FXR) and its downstream target *NROB2*

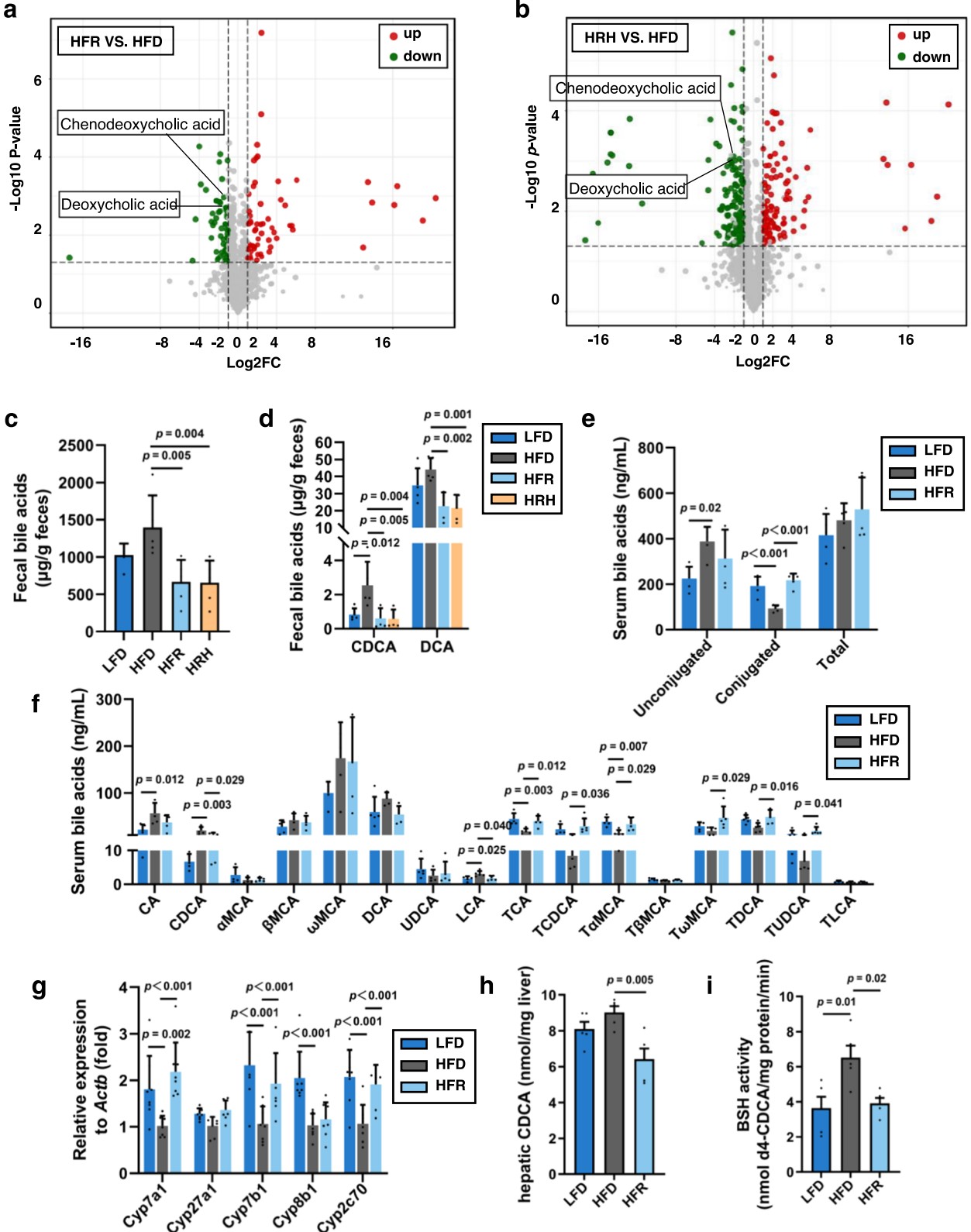

**Fig. 7 | REV-I attenuates HFD-induced fecal CDCA elevation. a, b** Volcano plots show differential metabolites (determined by Log2FC and −Log10 p value) in designated FEs. CDCA and DCA were annotated in the plot for both panels; n = 5. LCA level is also reduced in HFR-FE and HRH-FE compared to HFD-FE but did not reach statistical significance. **c** Fecal total bile acid levels. **d** Fecal CDCA and DCA levels. HRH, heated feces from HFR mice. **e** Serum levels of unconjugated, taurine-conjugated, and total bile acids. **f** Serum levels of major unconjugated and taurine-conjugated bile acids in mice. **g** Expression levels of genes that encode enzymes involved in bile acid biosynthesis in the liver; n = 8. **h** The concentration of CDCA in the liver. **i** The activity of BSH in feces is defined by the production of d4-CDCA per mg protein per min. n = 5 for each test. Statistical significance was evaluated by two-sided one-way ANOVA with Dunnett's post hoc test compared with the HFD group. See also Supplementary Figs. 6 and 7. Data are presented as mean ± SD. Source data are provided as a Source Data file.

(which encodes human small heterodimer partner, SHP) in Caco-2 cells. Importantly, they also activated *RELA*, *IL6* and *SCARB1* (Fig. 8a). Controversy exists on the effect of CDCA/FXR on NF-κB[30–32]. To explore the relationship among FXR, NF-κB and SR-B1, chemical inhibitors of NF-κB (QNZ) and FXR (Guggulsterone, Gug, a phyto-steroid found in the resin of the guggul plant[33]) were utilized. Gug addition blocked CDCA-induced expression of *SCARB1*, *NR1H4*, *NROB2 and FGF19*, while QNZ addition generated no such blockage (Fig. 8b, c). Furthermore, Gug treatment activated *RELA* on its own, and CDCA can stimulate *RELA*, regardless of absence or presence of QNZ or Gug. In the presence of QNZ or Gug, stimulatory effect of CDCA on *Il6* and *Il1b* was absent (Fig. 8d). While the complicated relationships among FXR, NF-κB and SR-B1 require intensive further investigations, we can conclude that FXR is required for CDCA to activate gut SR-B1.

We then assessed jejunal expression of FXR signaling related genes in mice on LFD, HFD or HFR for 8 weeks. Although HFD did not increase expression of *Nr1h4*, REV-I repressed its expression (Fig. 8e). HFD feeding increased jejunal levels of *NrOb2* and *Fgf15*, while elevated *Fgf15* expression was significantly attenuated by REV-I. For the other two FXR-related genes (*Slc51a* and *Slc51b*), no appreciable difference was noted (Fig. 8e). These two genes encode the heteromeric organic solute transporter alpha/beta, responsible for exporting BA across the basolateral membrane to systemic circulation. The expression of these two genes is induced mostly by bile flux in ileum[34]. It remains to be determined whether the window of regulating their expression at mRNA level in jejunum by HFD challenge and REV-I is relatively narrow. Finally, we included GW4064 into another set male mouse study. Six-week GW4064 treatment further increased postprandial TG, TRL-TG and TRL-ApoB48 levels in HFD fed mice, and the treatment also blocked the inhibitory effect of REV-I (Fig. 8f–h). Furthermore, GW4064 treatment induced jejunal but not hepatic SR-B1 expression in HFD-fed mice and blocked the attenuating effect of REV-I on jejunal SR-B1 expression (Fig. 8i).

## Discussion

Although numerous studies have shown profound effects of dietary polyphenol interventions on improving insulin signaling and attenuating dyslipidemia; for decades, mechanisms underlying these beneficial effects remain elusive. More and more attention has been made to contributions of gut microbiome toward metabolic homeostasis[2,4,9,11,35]. Bringing gut microbiome and gut metabolomics into mechanistic understating of dietary polyphenol intervention will help us to resolve the puzzle of why those interventions target multiple organs with extremely low bioavailability. We show here that inhibition of chylomicron secretion is a previously unrecognized function of REV-I, involving gut specific SR-B1 repression. Moreover, we observed the effect of REV-I on attenuating HFD-induced fecal BA elevation. Utilizing CDCA, the most potent natural FXR agonist, as an example, we demonstrated its stimulation on SR-B1. We hence suggest that REV-I targets gut microbiome, leading to the regulation of BA homeostasis and FXR activity, and the attenuation of SR-B1-mediated chylomicron secretion.

The high concentration of resveratrol in red wine explains the "French paradox," referring to the fact that French people have lower rates of CVD although they consume more saturated fat-rich diets. In overweight and obese human subjects, high-dose (1–2 grams daily) resveratrol reduced both intestinal and hepatic lipoprotein particle production[36]. ApoB48 production rate in those subjects receiving 2-week resveratrol treatment was reduced by 22%[36]. In this clinical investigation, potential involvement of gut SR-B1 inhibition was not assessed.

Approximately 50,000 PubMed publications have been generated to date on three major dietary polyphenols. A common feature of those plant-derived "nutraceuticals" is the lack of defined receptors. Those dietary polyphenols were shown to target multiple organs and regulate multiple signaling cascades. This feature is shared by known drugs including digoxin, acetylsalicylic acid, artemisinin, and metformin[37]. Such a feature also makes the journey for the mechanistic understanding of their biological functions relatively lengthy.

SR-B1 is widely expressed in metabolic active tissues such as the liver, adrenal gland, intestines, and vascular endothelium[38–40]. In the liver, it serves as the HDL receptor and was suggested to mediate reverse cholesterol transport to prevent atherosclerosis[41]. In vascular endothelial cells, it drives LDL transcytosis to promote atherosclerosis[42]. In small intestine, we reported that SR-B1 expression was the highest in the jejunum, and jejunal SR-B1 over-expression led to chylomicron over-production[22]. We suggest that it is necessary to bring tissue-specific transgenic mouse models into our investigations for dissecting their functions in each cell lineage. In the current study, we showed that 8-week REV-I prevented HFD-induced jejunal SR-B1 over-expression and reduced chylomicron secretion. Utilizing intestinal mucosa-specific SR-B1 KO mouse model and the chemical inhibitor BLT-1, we verified that targeting jejunal SR-B1 is a key mechanism for REV-I in attenuating chylomicron secretion.

More and more attention has been made to the role of gut microbiome in regulating metabolic homeostasis[3,6,9,10], including its role in mediating the beneficial effects of REV-I[4,10,11]. Mechanistic insights on the involvement, however, need to be explored. There are several ways that interactions among dietary polyphenols, gut microbiome, and ingested food, can influence metabolic homeostasis. Firstly, gut microbiota metabolites of a polyphenol may exert certain functions. In studying anthocyanin, we observed that its microbial metabolite protocatechuic acid possesses remarkable anti-atherogenic effects[8]. Secondly, dietary polyphenols may reverse gut microbial dysbiosis caused by HFD[10]. Gut microbiota may also produce certain products with metabolic functions. Administration of *Akkermensia muciniphila* (*A. muciniphila*) or its out-membrane protein Amun_1100 was shown to improve metabolic homeostasis or the gut barrier[43]. Furthermore, the drastic expansion of *A. muciniphila* was observed in HFD-fed mice with intervention utilizing polyphenol-rich extract from cranberry[44]. Finally, dietary polyphenol intervention may affect fermentation processes of gut microbiota on ingested food. It is well known that short-chain FAs (SCFAs), derived from intestinal fermentation, exert multiple functions in the gut and elsewhere[45].

We observed the general beneficial effect of REV-I on improving gut microbiota composition, reported in previous studies[4,10]. It is also worth mentioning that although previous studies suggested that a product of *A. muciniphila*, which belongs to the phylum *Verrucomicrobia*, possesses metabolic beneficial functions[43], REV-I did not increase the genus of *Akkermansia* in our study. Decreased relative abundance of *Akkermansia* was also observed by Sung et al. in conducting their resveratrol study in mice[4]. Further investigations are needed to expand the investigation into bacterial species levels. We then proved the anti-inflammatory feature of a major microbial metabolite of resveratrol, DHR. This metabolite, as well as 4-hydroxyflavone, possesses anti-inflammatory features in our in vitro settings. Nevertheless, they are unlikely the entities in mouse FEs that directly regulate SR-B1.

Among hundreds of compounds identified via metabolomics profiling, we picked BAs for further study as recent investigations reveal that the crosstalk between BAs and gut microbiome greatly impacts host metabolism[9,35,46–49]. BAs are among critical components of the gastrointestinal tract that link gut microbiome to metabolism as well as intestinal permeability and inflammation[27]. A recent study showed that theabrownin from Pu-erh tea exerts its effect on attenuating hypercholesterolemia via increasing the level of gut conjugated BAs, involving the suppression of BSH-producing gut microbiome[48]. Furthermore, BAs including CDCA are heat stable, which has been verified in our current study (Fig. 7c, d).

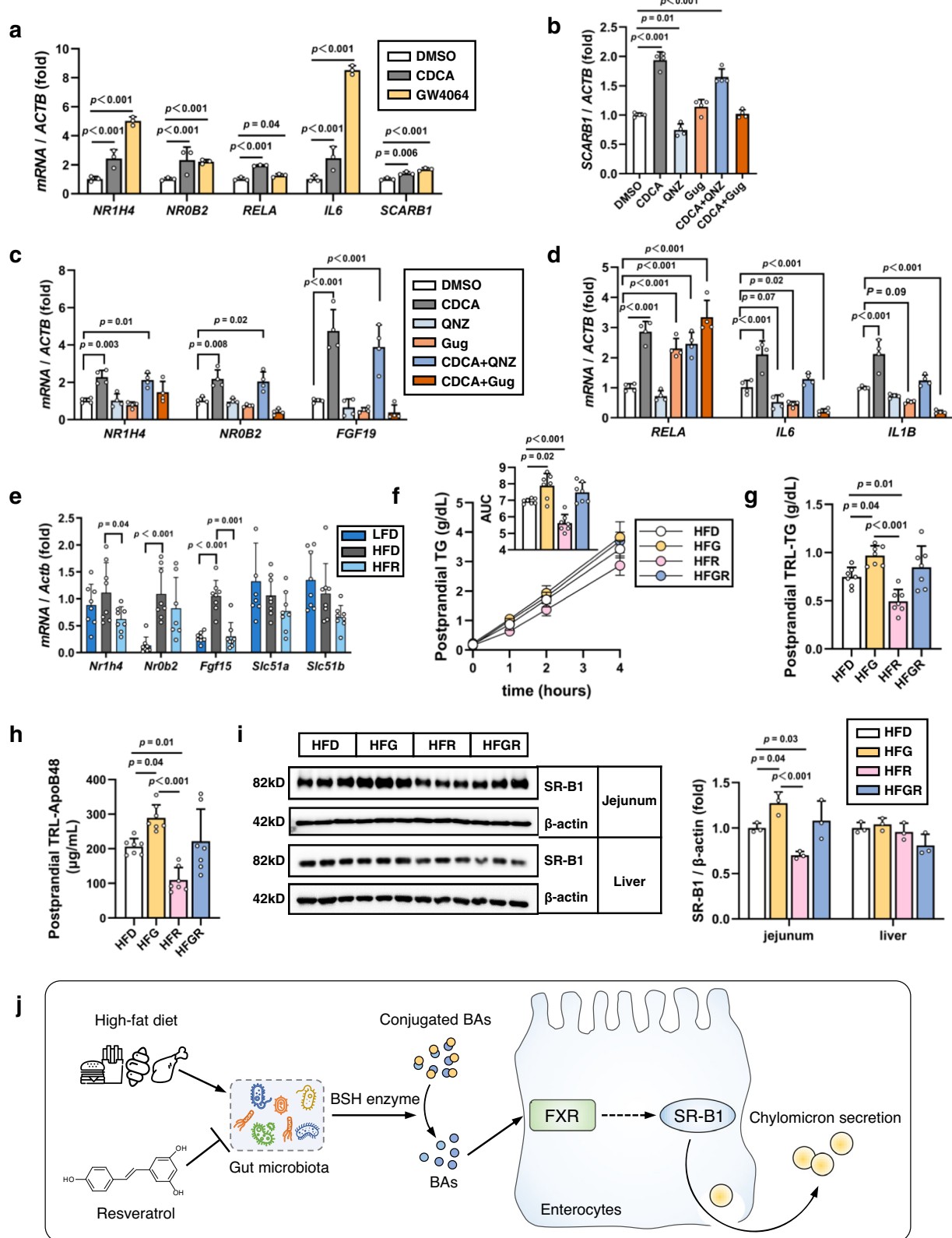

**Fig. 8 | CDCA and FXR agonist upregulate SR-B1 while FXR agonist administration blocks function of REV-I. a** Relative expression levels of *NR1H4* (which encodes FXR), *NROB2* (which encodes SHP), *SCARB1*, *RELA*, and *IL6* in Caco-2 cells treated with CDCA (200 μM), or GW4064 (25 μM) for 8 h; *n* = 3. **b**–**d** The relative expression levels of *SCARB1*, *NR1H4*, *NROB2*, *FGF19*, *RELA*, *IL6* and *IL1B* in Caco-2 cells treated with CDCA, or indicated inhibitor, or CDCA plus indicated inhibitor for 24 h. QNZ, NF-κB inhibitor (100 nM); guggulsterone (Gug), FXR antagonist (20 μM); *n* = 4. **e** The relative expression level of genes in the FXR signaling pathway detected by qRT-PCR in jejunum of designated mouse groups; *n* = 8. **f** Postprandial TG levels

and AUC during FTT in mice treated with HFD, HFR, HFD + GW4064 (HFG) or HFR + GW4064 (HFGR) for 6 weeks. **g** Postprandial TG levels in TRL. **h** Postprandial ApoB48 levels in TRL. **i** Jejunal SR-B1 level in designated mouse groups; *n* = 7 for above tests. **j** Diagram shows the effect of HFD challenge and REV-I on fecal bile acid level, involving bacterial produced BSH; as well as their effect on SR-B1 mediated chylomicron secretion, dependent on FXR. Statistical significance was evaluated by two-sided one-way ANOVA with Dunnett's post hoc test compared with the DMSO group or HFD group. *$p < 0.05$, **$p < 0.01$, ***$p < 0.001$. Data are presented as mean ± SD. Source data are provided as a Source Data file.

Primary BAs such as CDCA can be converted into secondary bile acid by microbial modifications. Bacterial produced BSH and the cytochrome P450 enzyme CYP2C70 participate in the homeostasis between conjugated and de-conjugated bile acids[27,28]. As the most potent natural agonist of FXR, CDCA level can be elevated after dietary challenge. Hepatic and gastrointestinal FXR, as the principal sensor of bile acids, play a fundamental role in the feedback regulation of BA synthesis. Briefly, excessive BAs contribute to hepatic FXR activation and the expression of SHP, which inhibits transactivation of genes that encode CYP7A1 and CYP8B1, two key enzymes for BA synthesis[50]. In the gut, FGF15/FGF19 can be induced by FXR, which may also result in transcriptional repression of CYP7A1[51].

Consistent with our observations, two previous studies have suggested that FXR activation upregulates SR-B1. GW4064 treatment or tail vein injection of adenovirus-FXR was shown to stimulate hepatic SR-B1, while such stimulation was absent in $FXR^{-/-}$ mice[52]. Oral gavage of CDCA was also shown to increase mouse hepatic SR-B1 levels[53]. Another study showed that SR-B1 expression was stimulated at mRNA and protein levels in HepG2 cells treated with GW4064 or CDCA, and an FXR binding element (FXRE) was located in human *SCARB1* promoter[29]. Controversy, however, exists on effect of CDCA/FXR signaling on NF-κB[30–32]. It has been reported that activation of FXR inhibited NF-κB and inflammatory responses in the liver[30,54], while other studies showed that intrahepatic accumulation of hydrophobic bile acids such as DCA and CDCA at μmol levels directly induced NF-κB activation[31,32]. We show here that CDCA can stimulate NF-κB in the presence of FXR inhibition, suggesting that the stimulation may involve a nuclear receptor other than FXR. Importantly, CDCA cannot stimulate SR-B1 when FXR was inhibited. While further investigations are needed to explore relationships among FXR, NF-κB and SR-B1 in complicated physiological and pathophysiological settings, we show that in vivo FXR activation blocked REV-I-dependent suppression of SR-B1.

In summary, we bring SR-B1 as well as FXR signaling into the understanding of metabolic beneficial effect of REV-I. As shown (Fig. 8j), HFD challenge impairs lipid homeostasis, including the stimulation of gut SR-B1 expression and elevated chylomicron secretion. This impairment is associated with elevated BAs in feces. Although we proved the stimulatory effect of CDCA on SR-B1 expression in Caco-2 cells, in vivo FXR/SR-B1 activation may involve other BAs. Different BAs may serve as either FXR agonist or antagonist of FXR. REV-I can target gut microbiome, improving gut microbiome diversity[4,10,11]. In addition, REV-I reduces the density of BSH-producing microbiomes, associated with reduced fecal BSH activity. This may contribute to reduced fecal bile acid levels, as BSH catalyzes the hydrolysis of conjugated bile salts into de-conjugated BAs, including CDCA. Although NF-κB has been identified as a trans-activator of jejunal SR-B1 in our current study (Fig. 5), and CDCA can activate *RELA* (which encodes NF-κB), the stimulatory effect of bile acid, such as CDCA, on SR-B1 requires FXR. As in vivo FXR activation blocked the effect of REV-I on repressing gut SR-B1, we suggest that FXR is among the critical targets of REV-I on lipid homeostasis.

Our investigation opens an avenue for mechanistic exploration of how dietary polyphenols exert their metabolic effects via interacting with gut microbiota, including the regulation of gut bile acid/FXR signaling cascade. Future directions include studies on involvement of other bile acids, as well as on further mechanistic exploration of how gut microbiome is involved in fecal bile acid homeostasis in response to REV-I, including their synthesis in the liver, and their fecal modifications and eliminations. Another one is to determine whether the identified FXR/SR-B1 signaling cascade applies to functions of other dietary polyphenols and nutraceuticals. Functions of other differential metabolites between HFR and HFD, identified in our metabolomics analyses, should also be further studied. Female mice are known to be resistant to HFD challenge, involving the participation of estradiol[55];

while resveratrol can act as a mixed agonist and antagonist for estrogen receptors[56]. Further knowledge advancement is required to expand our investigations into female rodent models.

## Methods

### Animal study

All animal protocols were approved by the University Health Network Animal Care Committee, or The Hospital for Sick Children Animal Care Committee, or the Animal Ethics Committee of Sun Yat-sen University. Six-week-old male C57BL/6J mice were purchased from Jackson Laboratory or Charles River Laboratories. $iScarb1^{-/-}$ mice were generated by mating floxed SR-B1 ($Scarb1^{fl/fl}$) mice[42] with Villin-Cre mice (Jackson Laboratory, Stock#021504). The method for creating $Scarb1^{fl/fl}$ mice has been described previously[42]. The primers utilized for genotyping were listed in Supplementary Table 2. Mice were group-housed in individually ventilated cages under a constant temperature (22 °C) and humidity (40–60%) with a 12-h light/dark cycle and ad libitum access to food and water (<or =5 mice per cage). Only male C57BL/6J mice were used because female mice are not susceptible to diet-induced obesity and metabolic disorders, such as insulin resistance and hypertriglyceridemia[55]. For the HFR diet, 100 g of 60% high-fat diet was smashed into powder-like small pieces before 0.5 g of resveratrol was scattered to the diet by several batches. The mixture was stirred thoroughly from the bottom to the top, made into pellets by compression molding and stored at −20 °C until use. The diet was served under a hand-made light-proof shelter.

In the REV-I study, 6-week-old male C57BL/6J mice were fed with low-fat diet (LFD) or high-fat diet (HFD), or HFD with resveratrol (0.5% in diet, designated as HFR) for 8 weeks. The mouse cecum content and fresh feces were collected for 16S rRNA profiling and metabolomics study, and serum was collected for BA analysis. In another set of mice experiment, 6-week-old male C57BL/6J mice were fed with LFD, HFD, or HFR (daily oral gavage of resveratrol at 500 mg/kg body weight) for 12 weeks, and mouse feces were collected for TG output measurement.

For REV-I in SR-B1 KO mice, 6-week-old male intestine mucosa-specific $Scarb1^{-/-}$ ($iScarb1^{-/-}$) mice or control $Scarb1^{fl/fl}$ littermates were fed with HFD without or with REV-I respectively ($Scarb1^{fl/fl}$ – R, $Scarb1^{fl/fl}$ + R, $iScarb1^{-/-}$ – R, $iScarb1^{-/-}$ + R groups) for 8 weeks.

For resveratrol and BLT-1 co-treatment study, 6-week-old male C57BL/6J mice were fed with LFD, HFD or HFR for the first 4 weeks. Then mice in the HFD and HFR groups were divided into two subgroups and subject to daily oral gavage of BLT-1 (Millipore, Cat#373210), a specific SR-B1 inhibitor, at 3 mg/kg body weight or solvent (PBS containing 0.5% DMSO) for 2 more weeks.

For fecal microbiota transplantation study, 6-week-old male C57BL/6J mice were fed with LFD or HFD for the first 4 weeks. Fecal microbiota transplantation (FMT) was then conducted as detailed below, followed by metabolic tolerance tests at indicated time.

For the FXR agonist administration study, 6-week-old male C57BL/6J mice were fed with HFD, HFR, HFD + GW4064 (Selleck, Cat#S2782) (the HFG group), or HFR + GW4064 (the HFGR group) for 6 weeks. GW4064 was dispersed in PBS containing 0.5% sodium carboxymethylcellulose before being sonicated for 10 min and was administered 5 times a week by oral gavage at 10 mg/kg body weight.

At the end of these experiments, 4–5 mice per group were selected randomly for fat tolerance test (FTT) as detailed below. Mice were fasted overnight (12 h) before being euthanized with $CO_2$ treatment followed by cervical dislocation, with protocols approved by University Health Network Animal Care Committee, or The Hospital for Sick Children Animal Care Committee, or the Animal Ethics Committee of Sun Yat-sen University. Blood samples were collected and centrifuged at 4 °C, $2292 \times g$ for 10 min to obtain serum samples. Tissues from liver and small intestine, cecum contents and feces were collected, put into liquid nitrogen immediately and then stored at −80 °C for further analysis.

## Caco-2 cell culture and treatment

Human Caco-2 cells (ATCC, Cat#HTB-37, isolated from colon tissue of a 72-year-old white male person) were cultured in Dulbecco's modified Eagle's medium (containing 4.5 g/l glucose) supplemented with 20% fetal bovine serum, 100 units/ml penicillin and 100 µg/ml streptomycin. Cells were grown at 37 °C in a 5% $CO_2$ humid atmosphere. To establish intestinal barrier model, as reported by Briand et al.[57], Caco-2 cells were grown for 21 days on microporous PET transwell inserts (23.1 mm, 1 µm pore size, Becton Dickinson Labware) (Supplementary Fig. 3d) until fully differentiated into a monolayer. After a 12-h starvation (serum-free) period, the monolayers were treated with lipid micelle (containing 1.6 mM oleic acid, 1 mM taurocholic acid sodium, 0.2 mM 1-monooleoylglycerol, 0.05 mM cholesterol, 0.2 mM L-α lyso-phosphatidylcholine and 0.02 mM BODIPY labeled $C_{12}$ fatty acid), without or with 25 µM resveratrol, or diluted sterile fecal extract (FE) from HFD or HFR-fed mice. For studying chylomicron secretion, media in the lower compartment was collected to measure the concentration of BODIPY labeled $C_{12}$ fatty acid and read at excitation wavelength of 475 nm and emission wavelength of 520 nm with a fluorescence spectrophotometer (Cytation 5, BioTek, USA).

## Metabolic tolerance test

Methods for oral glucose tolerance test (OGTT), intraperitoneal insulin tolerance test (IPITT), and FTT have been described previously[24,58]. For OGTT and IPITT, mice were fasted in the morning for 5 h. Blood glucose levels were determined at 0, 15, 30, 60, 90 and 120 min after oral gavage of glucose at 2 g/kg body weight or intraperitoneal injection of insulin at 0.5 U/kg body weight. To study intestinal chylomicron secretion, we performed fat tolerance test in which mice were fasted overnight prior to oral gavage of olive oil (Sigma-Aldrich, Cat#75348) at 10 ml/kg body weight, and intraperitoneal injection of poloxamer 407 (Sigma-Aldrich, Cat#P2443) at 1 g/kg body weight to block lipolysis. Blood was then collected from the tail vein at 0, 1, 2 and 4 h for triglycerides (TG) detection. Triglyceride-rich lipoprotein (TRL) fraction (mainly chylomicron in this context) was isolated as detailed below from mouse plasma collected at the 4th h of FTT. TRL-TG concentrations were detected by a commercial assay kit (Sigma-Aldrich, Cat#TR0100) and TRL-ApoB48 protein levels were measured by western blotting. To determine TG produced by the liver, mice fasted overnight were injected intraperitoneally with poloxamer 407 at 1 g/kg body weight, and blood was collected from the tail vein at 0, 1, 2 and 4 h to measure TG levels. Then TRL was isolated and TRL-TG and TRL-ApoB48 levels were measured as stated above.

## TRL fraction isolation

Mouse plasma was obtained by centrifuging blood at 5860 × g for 15 min at 4 °C. In total, 150 µl of plasma was used to isolate TRL as we described previously[59]. In brief, 150 µl of plasma was added to the bottom of a centrifugation tube (Beckman Coulter, Cat#344057) containing 4 ml of potassium bromide solution with density of 1.006 g/cm³. Then the sample was centrifuged in an ultracentrifuge (Beckman Coulter, Optima XE-100) at 116,140 × g for 70 min at 10 °C using a SW55Ti rotor (Beckman Coulter). The top layer of 300 µl was collected as the TRL fraction which was used for measuring TG concentration and ApoB48 level.

## Post-heparin plasma LPL activity measurement

LPL is located at the luminal side of capillaries and arteries where it hydrolyzes TG in chylomicron or VLDL to produce free fatty acids. By injection of heparin into the vein, the anchor between LPL and epithelial cells was cleaved and LPL was thus released to circulation in which we can measure its activity by adding an exogenous substrate. In our study, fasted mice were injected through the tail vein with heparin sodium (BioShop, Cat#HPA333.25) at 300 U/kg body weight.

Blood was collected 10 min later and centrifuged at 5860 × g for 15 min at 4 °C to obtain mouse plasma[60]. Then the plasma was used for measuring LPL activity with a commercial fluorescence assay kit (Abcam, Cat#ab204721).

## TG intake and fecal TG output measurements

In animal experiment 2, after 11 weeks of HFD feeding, mice were housed individually in metabolic cages. Food intake was recorded daily, and TG intake was calculated based on fat content of the diet. Fecal samples were collected daily for three consecutive days. Then feces were lyophilized, ground, and stored at −80 °C before analysis. Fecal TG was measured by an enzymatic assay with a commercial kit (Applygen, Cat#E1013).

## Fecal microbiome transplantation (FMT)

Fresh feces were collected from wild-type C57BL/6J mice (donor mice) fed with HFD or HFR for 8 weeks. Feces were collected, homogenized in PBS with 0.05% cysteine HCl and then filtered through a 100 µm strainer as described by Sung et al.[4]. Six-week-old male C57BL/6J mice (recipients) on LFD or HFD for 4 weeks were administered with PEG3350 at 17 mmol/l in water and fasted overnight to eliminate their gut microbiome before the first FMT. Each mouse then received designated FMT (200 µl, about 40 mg of freshly collected feces) as indicated in Fig. 6a for a total of three times on days 1, 3 and 5. Both LFD and HFD-fed recipients received one of the three designated FMT. Mice received fecal slurry from HFD-fed mice were designated as HFD-FMT group, mice received HFD and resveratrol-fed mice were designated as HFR-FMT group. Another sub-group designated as HFD-FMT + R were included, in which mice received HFD-FMT along with oral gavage of resveratrol at 500 mg/kg body weight on day 1, 3 and 5.

## Histological analysis

Oil-red O staining of liver samples was conducted as in our previous study[61]. Fresh mouse liver was embedded in OCT compound (Sakura, Cat#4583) and cut into 8-µm sections. After being fixed in 4% paraformaldehyde for 30 min, the sections were subject to Oil-Red O staining (Sigma-Aldrich, Cat#O0625) dissolved in 60% isopropanol for another 30 min. Then the sections were differentiated in 60% isopropanol and counterstained with Harris hematoxylin. Jejunum samples were fixed in 4% paraformaldehyde for 24 h and then paraffin-embedded, dewaxed in xylene, rinsed in alcohol, rehydrated, and unmasked in citrate buffer (PH 6.0). After block with protein block solution (Dako, Cat#X0909), sections were incubated with SR-B1 antibody (Abcam, Cat#ab217318, 1:500) for 45 min at 25 °C, followed by incubation in Alexa Fluor 555 conjugated anti-rabbit IgG (Thermo Fisher Scientific, Cat#A-21428, 1:200) for 30 min at 25 °C. Then cell nuclei were counterstained with DAPI, and anti-fade mounting media was applied before immunofluorescence analysis. Alternatively, sections were incubated in ImmPress anti-rabbit IgG (Novus Biologicals, Cat#MP-7401-NB, 1:200) for 30 min at 25 °C after incubation in SR-B1 primary antibody, developed with DAB substrate and counterstained with Harris hematoxylin for immunohistochemical analysis.

## Sterile FE preparation

Two freshly collected fecal pellets (about 40 mg) from mice fed with HFD or HFR for 8 weeks were collected and homogenized in 500 µl PBS. Following a sonication procedure on ice for 1 min twice, and a filtration procedure (through a 0.22 µm strainer), sterile fecal FE was obtained. Then half of the FE from HFR group was heated at 95 °C for 5 min and denoted as the HRH group. The absence of alive bacteria in sterile FE was verified by applying 10 µl FE on Luria broth (LB) agar plate. Thereafter, these sterile FE was diluted and applied to treat Caco-2 cells or intestinal barrier models or subject to metabolomic profiling.

## Microbiome profiling and data analyses

DNA was extracted from mouse cecum content or feces using the Qiagen PowerSoil Pro Kit. In brief, the V4 hypervariable region of the 16rRNA gene was amplified using barcoded 515F and 806R primers. Paired end (150 bp) sequencing was performed on an Illumina MiSeq platform[62]. The UNOISE pipeline as implemented through USEARCH was used to process raw sequence reads[63]. Cutadapt v.2.6 was used to trim the last base from all reads. Reads were quality filtered and paired ends reads were assembled to a minimum or maximum length of 243 and 263 (±10 from the mean) base pairs. Next singletons and chimeras were removed. A 99% identity was used to identify operational taxonomic units (OTUs). SINTAX was used to assign taxonomy using USEARCH and the Ribosomal Database Project database v18 available through UNOISE[64]. QIIME1 was used to align OTUs.

OTU tables were rarified to an even depth (12,403 reads). Samples were proportionally normalized by dividing read counts by the total number of reads prior to beta-diversity analyses. Bray–Curtis dissimilarity was used to assess community (beta-diversity) differences between groups. Permutational multivariate analyses of variance (PERMANOVAs) were conducted to assess Bray–Curtis beta-diversity differences. Principal coordinate analysis (PCoA) plot was used to visualize beta-diversity differences. The phylum level differences of Bacteroidetes and Firmicutes were assessed using Student's $t$ test and Mann–Whitney nonparametric test and visualized using PRISM. Genus level differences were assessed using Wald test as implemented in DESeq2 and visualized using PRISM. $p$ values were adjusted for multiple tests using Benjamini–Hochberg correction.

## Metabolomics profiling and data analyses

Sterile FE was prepared and stored at −80 °C as stated above. Samples were thawed on ice, sonicated for 10 min and centrifuged twice at 13,200 × $g$ for 10 min to discard insoluble particles. The supernatant was collected for LC-ESI-MS/MS analysis. The untargeted metabolomics analysis was performed by Metware Biotechnology Co., Ltd (Wuhan, China) using a UPLC (ExionLC AD system, AB SCIEX) coupled with a Quadrupole-Time of Flight mass spectrometer (TripleTOF 6600, AB SCIEX). The widely targeted metabolomics profiling was performed using a UPLC coupled to a high-resolution tandem mass spectrometer (QTRAP®, AB SCIEX) controlled by Analyst 1.6.3 software (SCIEX). Samples were separated over an ACQUITY HSS T3 (2.1 × 100 mm, 1.8 μm) column at 40 °C. All samples were equivalently mixed to make quality control (QC) samples and analyzed with other samples. A volume of 5 μl of FE was injected into the UPLC-MS. Mobile phases consisted of (A) LC-MS grade water with 0.1% formic acid and (B) acetonitrile with 0.1% formic acid with a flow rate set to 0.4 ml/min. The following gradient was applied: A/B (95/5%) at 0 min, then B to 90% from 0 to 10 min, then B held at 90% from 10 to 11 min, and finally changed to A/B (95/5%) at 11.1 min and held to 14 min. The MS was operated in both positive and negative modes.

Empirical cumulative distribution function was applied to analyze the frequency of QC samples with coefficient of variation (CV) value less than 15%. More than 80% of QC samples have a CV value <15%, indicating a reliable dataset[65,66]. All raw data was normalized using Quantile normalization and then Log2 transformation was performed before analyses. For principal component analysis (PCA), unit variance scaled data was analyzed by statistics function prcomp within R v.3.5.1. Differential metabolites were determined by false discovery rate (FDR) < 0.05 and absolute Log2FC (fold change) ≥ 1. To study the relative abundance of metabolites in different groups, differential metabolites were scaled using unit variance scaling method and then clustered by K-Means method into different subclasses. For pathway enrichment analysis, the KEGG database (http://www.genome.Jp/kegg/) was used to find enriched metabolic signaling pathways involving differential metabolites between two groups.

## BA analysis

Fresh fecal samples (~20 mg) were grinded with ball mill and extracted with 200 μl of methanol/acetonitrile (v/v = 2:8). Ten μl of internal standard mixed solution (1 μg/ml) was added into the extract. Following a precipitation procedure at −20 °C for 10 min, samples were centrifuged at 4 °C, 13,200 × $g$ for 10 min. Supernatants were collected and evaporated to dryness, reconstituted in 100 μl of 50% methanol (v/v) for further LC-MS analysis. BA contents were detected based on the AB Sciex QTRAP 6500 LC-ESI-MS/MS platform. The analytical conditions were as follows: HPLC column, Waters ACQUITY UPLC HSS T3 C18 (100 mm × 2.1 mm i.d., 1.8 μm); solvent system, water with 0.01% acetic acid and 5 mmol/l ammonium acetate (A), acetonitrile with 0.01% acetic acid (B); The gradient was optimized at 5–40% B in 0.5 min, then increased to 50% B in 4 min, then increased to 75% B in 3 min, and then 75–95% in 2.5 min, washed with 95% B for 2 min, finally ramped back to 5% B (12–14 min); flow rate, 0.35 ml/min; temperature, 40 °C; injection volume, 1 μl. Bile acids were analyzed using scheduled multiple reaction monitoring (MRM). Data acquisitions were performed using Analyst 1.6.3 software (Sciex). Multiquant 3.0.3 software (Sciex) was used to quantify all metabolites.

## BSH activity measurement

Fresh fecal samples (about 50 mg) were dispersed in 250 μl of PBS and homogenized twice for 15 s at 60 Hz. Samples were then sonicated for 90 s with a 30 s interval in an ice bath before being centrifuged at 4 °C, 13,200 × $g$ for 30 min. The supernatant was transferred to a new tube for detection of protein concentration with a BCA Protein Assay Kit (Pierce, Cat#23227). The supernatant was diluted to 2 mg protein/ml by PBS. Then 10 μl of the sample solution was incubated with 180 μl of sodium acetate buffer (pH 5.2) and 10 μl d4-TCDCA (2 mM) at 37 °C for 30 min, and the reactions were terminated by plunging the samples into dry ice. In total, 100 μl of methanol was then added to the mixture, the samples were vortexed and centrifuged at 4 °C, 13,200 × $g$ for 20 min. The supernatants were transferred to sampling vials for LC-MS analysis. The activity of BSH was defined by the production of d4-CDCA per mg protein per min.

## Chromatin immunoprecipitation (ChIP)

NF-κB-p65 binding motifs in human *SCARB1* or rat and mouse *Scarb1* promoters were analyzed through PROMO (version 8.3, http://alggen.lsi.upc.es/). About 60 mg of frozen mouse jejunum samples were ground in liquid nitrogen and kept on dry ice all the time. Samples were cross-linked in PBS containing 1.5% formaldehyde with tubes rotating at room temperature for 15 min and 0.125 M glycine solution was added to stop the cross-linking reaction. Then samples were centrifuged and washed with ice-cold PBS twice before homogenizing in PBS until unicellular suspension was obtained. The cells were centrifuged and resuspended in lysis buffer (750 μl per 1 × 10⁷ cells) supplemented with 10 μl/ml PMSF, 1 μl/ml aprotinin and 1 μl/ml leupeptin. Following a sonication procedure to shear DNA to an average fragment size of 200–1000 bp, the lysate was centrifuged, and supernatant was transferred to a new tube. Ten μl of chromatin was removed as input. Two μg of NF-κB-p65 antibody (Cell Signaling Technology, Cat#8242, 1:100) or Rpb (RNA polymerase II) antibody (Cell Signaling Technology, Cat#2629, 1:50) was added into ~25 μg of DNA for one immunoprecipitation and the complex was incubated with rocking overnight. Ten μl of blocked staph A was added and incubated for 15 min at room temperature before 150 μl of elution buffer was added to release the chromatin-protein complex. Finally, chromatin was reversed and purified for PCR or real-time PCR using primers listed in Supplementary Table 3.

## Reverse transcription and real-time PCR

Total RNA was extracted from mouse jejunum and Caco-2 cells with Trizol reagent (Invitrogen). Reverse transcription was conducted with

a commercial kit (Applied Biosystems, Cat#4368814) and real-time PCR was performed on a real-time PCR system (Applied Biosystems, 7500) using a one-step SYBR Green master mix (Bioline, Cat#BIO-73001) according to manufacturer's protocols. The primers used for amplification of each gene are shown in Supplementary Table 1. At last, the relative expression level of each gene was calculated using the $2^{-\Delta\Delta Ct}$ method normalized by the expression of *Actb* or *ACTB* (which encodes mouse or human β-actin, respectively).

## Protein extraction and western blotting

Total cell lysates were obtained by homogenizing mouse jejunum, liver or Caco-2 cell samples in lysis buffer supplemented with protease inhibitors, then 30 μg of boiled lysates were separated by 5–8% SDS-PAGE and blotted onto a nitrocellulose filter membrane (0.45 μm, Bio-Rad). After blocking with 7% milk, blots were incubated with anti-SR-B1 (Abcam, Cat#ab217318, 1:3000), anti-ApoB (Midland Bioproducts, Cat#MBC-APB-G1, 1:1000), anti-alpha-Albumin (Nittobo America, Cat#MBC-ALB-G1, 1:1000), anti-NF-κB-p65 (Cell Signaling Technology, Cat#8242, 1:1000), anti-GAPDH (Cell Signaling Technology, Cat#2118, 1:1000), anti-β-actin (Cell Signaling Technology, Cat#3700, 1:1000), anti-alpha-tubulin (Cell Signaling Technology, Cat#2144, 1:1000) and other antibodies, followed by HRP-conjugated horse anti-mouse IgG (Cell Signaling Technology, Cat#7076, 1:2000), goat anti-rabbit IgG (Cell Signaling Technology, Cat#7074, 1:2000) or donkey anti-goat IgG (Santa Cruz Biotechnology, Cat#sc-2020, 1:2000) where appropriate. The bands were exposed with an enhanced chemiluminescence kit (Thermo Fisher Scientific, Cat#34580) and semi-quantified using ImageJ software (version 1.8.0, https://imagej.nih.gov/ij/) normalized by GAPDH, β-actin or alpha-tubulin bands. When needed, blots were stripped with Tris-HCl buffer containing 10% sodium dodecyl sulfate and 0.75% β-mercaptoethanol and incubated at 55 °C for 25 min before re-probed with another primary antibody.

## Statistical analysis

Results are presented as mean ± standard deviation (SD). Comparisons between groups were performed using an unpaired two-tailed *t*-test, Mann–Whitney nonparametric test (between two groups), one-way analysis of variance (ANOVA) followed by the Dunnett's post hoc test (compared with HFD or PBS group), or two-way ANOVA (for mice of different genetic background and diet) followed by Sidak's post hoc test where appropriate. Statistical details are included in the figure legends in which the *n* number means biological replicates in all experiments. Statistical analyses were performed with GraphPad Prism 8.0 (GraphPad Software, La Jolla, CA, USA). *, #, or &, $p < 0.05$; **, ##, or &&, $p < 0.01$; ***, ###, or &&&, $p < 0.001$.

## Reporting summary

Further information on research design is available in the Nature Portfolio Reporting Summary linked to this article.

## Data availability

The 16S rRNA sequencing raw data have been deposited in Sequence Read Archive (SRA) with the accession number PRJNA859433. The untargeted metabolomics profiling processed data that support the findings of this study have been deposited in Figshare with the DOI number https://doi.org/10.6084/m9.figshare.20325402.v1 and has also been provided as Supplementary Data 1–4 with this paper. Meanwhile, the untargeted metabolomics profiling raw data have been deposited in MetaboLights with the accession number MTBLS7654. Source data are provided with this paper.

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

## Acknowledgements

This study is supported by Canadian Institutes of Health Research (PJT159735 to T.J. and Foundation Grant #353989 to K.A.), the National Natural Science Foundation of China (grant number 81730090 and 81973022 to W.L.), and National Institutes of Health operating grant (HL131597 to P.W.S.). J.P. is a visiting Ph.D. student supported by China Scholarship Council. F.R. is supported by Canadian Institutes of Health Research doctoral award. J.N.F. is supported by the Banting & Best Diabetes Centre (BBDC)-Novo Nordisk Studentship and Canada Graduate Scholarships—Master's program (CGS M). We thank Drs. Gary Lewis and Herbert Gaisano (University of Toronto), and Ms. Shannon Steel (Hospital for Sick Children) for data discussion, manuscript editing, and suggestions. We thank iSlide (https://www.islide.cc/) for the access of public icons in making diagrams in this manuscript.

## Author contributions

Conceptualization, methodology: J.P. and F.R. Formal analysis: J.P. Writing—original draft: J.P. Validation: F.R. and W.S. Software and visualization: A.A.H. Investigation: J.P., F.R., A.A.H., W.S., D.L., J.G. and J.N.F. Resources: C.M. and P.W.S. Supervision and funding acquisition: X.Q., B.C., K.A., W.L. and T.J. Data curation, project administration, and writing—review and editing: K.A., W.L. and T.J.

## Competing interests

The authors declare no competing interests.
