## [Peer Review File · Nature Communications]

Resveratrol intervention attenuates chylomicron secretion via repressing intestinal FXR-induced expression of scavenger receptor SR-B1REVIEWER COMMENTS

Reviewer #1 (Remarks to the Author):

In this report, Pang et al. investigate the potential mechanisms of the metabolic beneficial effects of resveratrol. By using a HFD feeding model, they show that resveratrol intervention (REV-I) inhibits chylomicron secretion by reducing jejunal SR-B1 expression. They further nicely confirm these observations using either an intestinal-mucosa-specific SR-B1^{-/-} mice or a SR-B1 inhibitor BLT-1 model. To identify the factor(s) that can reduce SR-B1 expression by REV-I, they focus on the changes of gut microbiota and the feces metabolites induced by REV-I because resveratrol and its direct metabolites have no significant effects on SR-B1 expression. The authors finally claim that bile acids, particularly the chenodeoxycholic acid (CDCA), is the mediator of REV-I. They conclude that REV-I improves lipid homeostasis via attenuating CDCA-induced SR-B1 expression elevation.

Major comments:

1. While the roles of gut SR-B1 in mediating the metabolic effects of REV-I are well established, the CDCA effects in this system are still unclear. Therefore, the title needs to be more specific to SR-B1 and exclude the CDCA.
2. It is still unknown how does REV-I lower the gut CDCA levels. Is it through the change of specific gut microbiota or change the liver bile acid synthesis? Metabolomics results in Supple Figure 7 show many metabolites with fold of changes much larger than CDCA. The rationale to pick up CDCA as a key metabolite to explain the REV-I effects is not convincing.
3. Similarly, it is unclear how CDCA can upregulate SR-B1 expression. Figure 5 shows NF- κ B may directly regulate SR-B1 transcription. However, there is no evidence that CDCA can activate NF- κ B to upregulate SR-B1 expression. If CDCA activates FXR to upregulate SR-B1 expression, is there FXRE in the promoter or enhancer region of SR-B1 gene?
4. Figure 7F model shows that CDCA activate both NF- κ B and FXR to upregulate SR-B1 expression. However, previous reports suggest that activation of FXR may inhibit NF- κ B signaling.
5. Figure 7E, HFD seems to strongly upregulate Nr0b2 and Fgf15 expression, which are downregulated by HFR. However, the other two FXR target genes (Slc51a and Slc51b) have no response to either HFD or HFR, indicating FXR-independent effects. How about the Scarb1 mRNA change?

Minor comments:

1. It would be good to provide more detail in the method how authors prepare the HFR diet.
2. In page 5 second paragraph, the full name of TRL needs to be provided earlier.
3. In Figure 1G, the postprandial TG level in HFR seems significantly lower than that in the LFD control group. This result needs explanation.

Reviewer #2 (Remarks to the Author):

1. The noteworthy results: the role of jejunum SR-B1-mediated chylomicrons in REV-mediated lipid homeostasis.

2. Some major issues need to be carefully addressed.

2.1 The analytical performance of the metabolomic approach.

It is recommended that QC samples should be located closely, and that 60% of the metabolites should have a CV value < 15% in the metabolomic data, with a more relaxed criterion (20%) for the metabolites near their limit of quantification. 1, 2, 3 It is a low requirement that 'More than 90% of QC samples have a CV value < 0.3'. Therefore, the distribution of QC samples in the score plot of PCA based on the information of all samples (including QC and analytical samples) should be provided to evaluate the analytical performance of the metabolomic approach. In addition, what is the proportion of metabolites with the CV value less than 15%, 20% and 30%, respectively, among all detected metabolites in QC samples?

References

1. Gika HG, Theodoridis GA, Wingate JE, Wilson ID. Within-day reproducibility of an HPLC-MS-Based method for metabonomic analysis: application to human urine. *J Proteome Res* 6, 3291-3303 (2007).
2. Begley P, et al. Development and performance of a gas chromatography-time-of-flight mass spectrometry analysis for large-scale nontargeted metabolomic studies of human serum. *Anal Chem* 81, 7038-7046 (2009).
3. FDA U. Guidance for industry: bioanalytical method validation. US Department of health and human services, US FDA. Center for Drug Evaluation and Research, Rockville, MD, USA, (2001).

2.2 Metabolite identification.

It is known that multiple ions would be produced from a metabolite after ionization. The authors pointed out that "Through LC-MS/MS analysis, we detected 2053 metabolites." In line 253. Is '2053' the number of metabolite or that of metabolite ions? If '2053' is the number of metabolites, all the detected metabolites should be listed in the supplementary material, as well as key parameters for the identification, including the retention time, theoretical accurate mass, measured accurate mass, and mass deviation. Besides, metabolites verified by the reference standards should be labeled. Changes in differential metabolites should be provided in the supplementary material, including fold changes, P value, and parameters for the identification mentioned above.

2.3 Figure 7.

Why fecal extracts from LFD were not analyzed using the metabolomic approach? Since the authors pointed out that 'Dietary resveratrol intervention improves lipid homeostasis via attenuating HFD-induced fecal chenodeoxycholic acid and jejunal SR-B1 elevation', the change in CDCA induced by HFD is essential. Therefore, alterations in fecal metabolites in HFD compared to LFD should be provided, where metabolomic data of the LFD group are essential. In addition, changes in fecal bile acids (including CDCA and DCA) from LFD, HFD, HFR to HRH should be clearly presented, which could be provided in the way as Figure 7D, and 7E.

2.4 Lines 90-290.

The authors tried to demonstrate the involvement of gut microbiota in REV-mediated lipid homeostasis via regulating jejunal SR-B1. Why microbial changes in jejunal contents were not detected? Why metabolomic alterations in jejunal contents were neither conducted? Instead, metabolome in feces and microbiome in cecal/fecal contents were determined. The function of the jejunum identified by the authors does not specifically correspond to changes in fecal metabolome and alterations in microbiome in fecal/cecal contents.

As a primary bile acid, CDCA is synthesized in the liver and further metabolized into secondary bile acid in the intestine. Therefore, the increase in fecal CDCA in HFD group is related to the increase in the synthesis in the liver and/or the decrease in intestinal microbial degradation. To prove the important role of CDCA elevation mediated by microbial metabolism, it is necessary to find the key microbial enzymes degrading CDCA and verify it through the regulation of related enzymes. There is still much work to be done in this work on the role of microbial metabolism and CDCA in REV-mediated lipid homeostasis via suppressing jejunal SR-B1 expression.

We sincerely thank the two Expert Reviewers for the advice and encouragement. Based on those
valuable comments and advice, we have conducted additional bench work and made our extensive
revisions to this manuscript accordingly. Briefly, we have made three sets of wet-bench and dry-
bench work to address the following three sets of main concerns.

**A) More bench work evidence on the role of fecal CDCA:** Reviewer 1 wrote: “*While the roles*
*of gut SR-B1 in mediating the metabolic effects of REV-I are well established, the CDCA effects in*
*this system are still unclear. Therefore, the title needs to be more specific to SR-B1 and exclude*
*the CDCA*”. We have carefully discussed this kind suggestion among the authors. We feel that it
is necessary to clarify the role of CDCA and its downstream signaling molecule FXR in our current
manuscript as their involvement further supports our overview on the role of gut microbiome as
well as gut metabolites within the feces, leading to jejunal specific SR-B1 repression. Hence, we
have measured mouse fecal and serum bile acid (BA) levels in LFD, HFD and HFR mice (**new**
**data in Figure 7C-F**). Briefly, HFD feeding increased levels of total BA, CDCA and DCA in
feces, while REV-I attenuated their increases (**Fig. 7C-D**). In serum, HFD increased levels of
unconjugated BAs (including CDCA) but repressed levels of conjugated BAs, while concomitant
REV-I reversed such changes (**Fig. 7E-F**). We also observed that REV-I reduced liver CDCA
content (**Fig. 7H**). More importantly, we found that fecal BSH activity was increased by HFD
feeding and reduced by REV-I (**Fig. 7I**). BSH, produced by certain gut bacterial genus, converts
conjugated BAs into unconjugated ones. Thus, REV-I can reduce CDCA production in the liver
and attenuate the conversion of TCDCA (conjugated one) into active CDCA. Following these
novel findings, we have made further data analyses on gut microbiome profiling. As shown, in
both mouse feces (**Fig. S7D**) and caecum content (**Fig. S7H**), densities of BSH-producing genus
were reduced by REV-I. For this sake, we request the permission to keep the term CDCA within
our manuscript title. To follow the Journal style (= or less 15 words), we have revised the
manuscript title as: “**Resveratrol intervention attenuates chylomicron secretion via repressing**
**fecal chenodeoxycholic acid and jejunal scavenger receptor SR-B1**”.

In conducting the above revisions, we have also addressed a main concern raised by Expert
Reviewer 2 for including LFD-fed mice to confirm CDCA changes, as well as other related
concerns (detailed in our point-by-point responses to Expert Reviewer 2). We have also included
further literature citation and discussion, including the one by Huang et al. in 2019, published in
Nature Communications (Ref. #48)¹.

**B) How can CDCA upregulate SR-B1 expression?** We admit that this is a tough and complicated
question for us to provide a complete answer at the current stage yet, although we have made our
great effort in advancing our knowledge on this. It is already known that as the most potent native
ligand, CDCA activates the nuclear receptor FXR. We found that in Caco-2 cells, both CDCA and
the synthetic FXR agonist GW4064 activated *NRIH4* (which encodes FXR), its target *SHP*
(encoded by *NROB2*), *SCARB1* (encodes SR-B1), *RELA* (encodes NF- κ B), and *IL6* (**Fig. 8A in**
**this new version**). We have also proved that NF- κ B is among the transcriptional activator of SR-
B1 by ChIP and qCHIP (**Fig. 5 in this new version**). Next question is the complicated relationship
among FXR, NF- κ B and SR-B1. Utilizing QNZ (an NF- κ B inhibitor) and Guggulsterone (Gug, an
FXR inhibitor), we did further study in the Caco-2 cell model with **new data** presented in **Fig. 8B-**
**D**. Briefly, the stimulatory effect of CDCA on *SCARB1* can be blocked by Gug but not by QNZ.
Thus, FXR is required for CDCA to stimulate SR-B1. However, CDCA stimulated *RELA* (NF-
κ B) regardless of the presence or absence of QNZ or Gug. Thus, CDCA can stimulate *RELA*

independent of FXR. It is unlikely that the inhibitor QNZ lost its effect in our experimental settings,
as in the presence of QNZ, although CDCA was still able to activate *RELA* (NF- κ B) expression,
CDCA did not activate the downstream targets of NF- κ B, *IL6* and *IL1B*. Based on literature, both
FXR and NF- κ B are known targets of CDCA (see our point-by-point responses in below). Since
the blockage of FXR did not block the stimulation of CDCA on NF- κ B, we suggest that CDCA
can activate NF- κ B directly (may involve nuclear receptors other than FXR). We also suggest that
FXR can directly activate SR-B1, based on literature as well as our observation that FXR inhibition
blocked the stimulation on SR-B1 by CDCA. The stimulatory effect of FXR activation on SR-B1
expression has been reported previously (**Antherosclerosis, 2010**)². However, if CDCA can
directly activate NF- κ B, and NF- κ B is the transcriptional activator of SR-B1, how can FXR
inhibition block the activation of CDCA on SR-B1? Is FXR a co-factor of NF- κ B on regulating
SR-B1? Instead of further pursuing this complicated question at this current stage, which needs
intensive ChIP, qChIP, systematic transcriptome analyses in various manipulated cell models
along with *in vivo* and *ex vivo* verifications with FXR KO mouse models, we decided to further
pursue the *in vivo* function of FXR activation. We found that in HFD fed mice, *in vivo* GW4064
administration for 6 weeks increased chylomicron secretion and increased jejunal SR-B1
expression. Importantly, GW4064 blocked the effect of REV-I on jejunal SR-B1 (**New data in**
**Fig. 8F-I**). We have also revised our summary diagram. As we did not assess the direct effect of
CDCA on NF- κ B and the effect of FXR on SR-B1 with further molecular biological tools in our
current study, we used dotted arrows in the indicated areas in the diagram (**Fig. 8J**).

**C) Additional data analyses on our metabolomics analyses.** Thanks for the advice from Expert
Reviewer 2. We have made our revisions to method description, presentation, and have included
further data analysed. Please see our point-by-point responses to those advice from Expert
Reviewer 2.

The current study utilized male mice only, mainly because female mice are resistant to HFD
challenge induced insulin resistance and impairment on lipid homeostasis. As reported by other
teams and by our team in 2019 (Plos Biol.)³, this involves the female hormone estradiol.
Resveratrol can act as agonist or antagonist of estrogen receptors (ERalpha/beta)⁴. Thus,
breakthroughs are demanded in technology and others before we can expand our investigations
into female rodents. We have discussed this at the end of our Discussion. In several random clinical
trials with resveratrol intervention, although both male and female patients were recruited, none
of the related studies (as far as we know) have assessed their result separately (on male vs female).
Since this is a pre-clinical study, we did not comment the above human gender issue in our
discussion, although we have cited and commented the most relevant human studies (such as the
one conducted by Dash and colleagues)⁵.

Based on the journal style and regulations, we have made some re-arrangement of the Figures,
along with new data in new Figures and supporting Figures. We hope that our revisions have
improved the overall quality of this manuscript. Below are our point-by-point responses to the two
expert reviewers.

**REVIEWER COMMENTS**

Reviewer #1 (Remarks to the Author):

*In this report, Pang et al. investigate the potential mechanisms of the metabolic beneficial effects*
*of resveratrol. By using a HFD feeding model, they show that resveratrol intervention (REV-I)*
*inhibits chylomicron secretion by reducing jejunal SR-B1 expression. They further nicely confirm*
*these observations using either an intestinal-mucosa-specific SR-B1^{-/-} mice or a SR-B1 inhibitor*
*BLT-1 model. To identify the factor(s) that can reduce SR-B1 expression by REV-I, they focus on*
*the changes of gut microbiota and the feces metabolites induced by REV-I because resveratrol and*
*its direct metabolites have no significant effects on SR-B1 expression. The authors finally claim*
*that bile acids, particularly the chenodeoxycholic acid (CDCA), is the mediator of REV-I. They*
*conclude that REV-I improves lipid homeostasis via attenuating CDCA-induced SR-B1 expression*
*elevation.*

***We thank Expert Reviewer 1 for the very nice summary and encouragement.***

*Major comments:*

*1. While the roles of gut SR-B1 in mediating the metabolic effects of REV-I are well established,*
*the CDCA effects in this system are still unclear. Therefore, the title needs to be more specific to*
*SR-B1 and exclude the CDCA.*

**Thanks for the advice.** We have now carefully addressed this critique above by conducting further
wet-bench and dry-bench work and request the permission to keep CDCA within the revised
manuscript title (see above lines 7-33). REV-I may reduce liver CDCA production and attenuate
CDCA de-conjugation in feces as we discussed along with our new data in Fig. 7C-I and Fig. S7D
and H. Two additional literature citations were made on these aspects^{6,7}.

*2. It is still unknown how does REV-I lower the gut CDCA levels. Is it through the change of*
*specific gut microbiota or change the liver bile acid synthesis? Metabolomics results in Supple*
*Figure 7 show many metabolites with fold of changes much larger than CDCA. The rationale to*
*pick up CDCA as a key metabolite to explain the REV-I effects is not convincing.*

**Thanks for the above comments.** As we have stated above, results from our new experiments
(Fig. 7C-I and Fig. S7D and S7H) allow us to suggest that REV-I reduces gut CDCA level via
attenuating its liver production/modification and its gut conversion from conjugated one into
unconjugated one, with the participation of bacterial produced BSH. We have made our revisions
in the abstract (page 2, lines 12-14), results (page 11, line 14 to page 12, line 22) and discussion
(page 17, lines 2-9; and page 18, lines 14-18) on these new findings.

We have picked CDCA for reasons that are more detailed in this revised version (page 11, lines
5-13). One important reason is that bile acids are heat stable. Our initial exploration (Fig. S4F in
this revised version) revealed the existence of heat stable materials in mouse feces that regulates
gut SR-B1 expression. If a candidate molecule up-regulates SB-B1, its level in feces should be
increased by HFD feeding and reduced by REV-I, and vice versa. CDCA and DCA fall into such
category while CDCA is the most potent FXR agonist. By the way, our new data in Fig.7C-D
indicate that total bile acid, CDCA and DCA are heat stable.

*3. Similarly, it is unclear how CDCA can upregulate SR-B1 expression. Figure 5 shows NF-κB*
*may directly regulate SR-B1 transcription. However, there is no evidence that CDCA can activate*

*NF-κB to upregulate SR-B1 expression. If CDCA activates FXR to upregulate SR-B1 expression,*
*is there FXRE in the promoter or enhancer region of SR-B1 gene?*

**Thanks for raising this important and intriguing question.** As stated above, we have made our
great effort, with the utilization of FXG agonist, chemical inhibitors of FXR and NF-κB in
addressing the complicated relationships among FXR, NF-κB and SR-B1 (**lines 34-66**). Although
we are still unable to provide a clear picture on the relationship among CDCA/FXR, NF-κB and
SR-B1 at the molecular levels, we have advanced our knowledge in the field, showing that FXR
is required for CDCA to activate SR-B1, and suggesting that CDCA may activate NF-κB via
nuclear receptor/s other than FXR. We also hypothesize that FXR may be a co-factor of NF-κB in
stimulating SR-B1 (this was not discussed in current manuscript). We have discussed our
observations above on our new data (Fig. 8B-D) (**page 13, lines 1-16**), and conducted further
mouse work to verify that FXR activation up-regulate SR-B1 in vivo (**Fig. 8F-I**). We have revised
the diagram and the figure legend (**Fig. 8J** in this revised version).

Several previous studies found that activation of FXR upregulates SR-B1 expression. For example,
after GW4064 was used to treat primary mouse hepatocytes or adenovirus-FXR injection to the
tail vein of mice, SR-B1 mRNA levels were increased⁸. In *FXR*^{-/-} mice, GW4064 treatment did not
increase SR-B1 expression⁸. Mice administered CDCA by oral gavage also showed higher level
of SR-B1 expression in the liver⁹. Furthermore, a study showed that SR-B1 expression was
stimulated at mRNA protein levels in HepG2 cells treated with GW4064 or CDCA². Meanwhile,
this team found that there was a FXRE in the promoter of SR-B1 gene with the binding motif -703
AGGCCA_{cg}ttctagAGCTCA -684². We have discussed these literatures in this revised version
(**page 17, line 19 to page 18, line 10**).

*4. Figure 7F model shows that CDCA activate both NF-κB and FXR to upregulate SR-B1*
*expression. However, previous reports suggest that activation of FXR may inhibit NF-κB*
*signaling.*

**Thanks for raising this important question.** In our view, controversy does exist in the field on
their relationships. Two studies reported that activation of FXR inhibited NF-κB signaling and
inflammatory responses in the liver^{10,11}. However, consistent with our current study, at least two
previous investigations suggested the opposite. They reported that intrahepatic accumulation of
hydrophobic bile acids such as DCA and CDCA at μmol levels directly induced NF-κB signaling
pathway in a PI3K or PKC dependent manner^{12,13}. As we have discussed above, upregulation on
SR-B1 by FXR activation has been reported by Chao and colleagues². We have cited the literature
on both sides in this revised version (**page 17, line 19 to page 18, line 10**).

*5. Figure 7E, HFD seems to strongly upregulate Nr0b2 and Fgf15 expression, which are*
*downregulated by HFR. However, the other two FXR target genes (Slc51a and Slc51b) have no*
*response to either HFD or HFR, indicating FXR-independent effects. How about the Scarb1 mRNA*
*change?*

**Thanks for pointing out this interesting issue.**
In this revised version, the above data are presented in **Fig. 8E**. We intend to suggest that this is
likely due to the existence of complicated feedback loops within the *in vivo* system. *Nr0b2* and

*Fgf15* are the most important and direct downstream targets of the BA/FXR signaling, which are
more sensitive to BA/FXR activation (even though *Nr1h4* mRNA level was not significantly
stimulated by HFD feeding in our current experimental settings, the effect on expression of these
two target genes are substantial).

*Slc51a* and *Slc51b* (also known as *OSTα-OSTβ*; *SLC51A-SLC51B*) encode the heteromeric organic
solute transporter alpha-beta, responsible for exporting BA across the basolateral membrane to the
circulation and facilitating enterohepatic circulation. Thus, the expression of these two genes is
induced mostly by bile flux in ileum¹⁴. We are not sure whether the *in vivo* window of regulating
their expression at mRNA level in mouse jejunum by BA/FXR activation in response to HFD
challenge and REV-I is relatively narrow. We intend to keep those *in vivo* observation data in this
manuscript, as the data may be interesting for some of our peers in their investigations.
Nevertheless, we made revision in the text (**page 13, line 20 to page 14, line 2**). As our focuses
are CDCA/FXR on lipid homeostasis and jejunal SR-B1, we conducted more *in vivo* activation of
CDCA/FXR, with new data presented in **Fig. 8F-I**.

**Yes**, the mouse jejunal *Scarb1* mRNA change data is presented in Fig. 2E, along with statistical
analysis presented in Table S4, which was increased by HFD feeding and repressed by REV-I.

*Minor comments:*

*1. It would be good to provide more detail in the method how authors prepare the HFR diet.*

**Thanks for the suggestion made by Expert Reviewer 1.** We have included detailed preparation
method in this revised version (**page 25, line 12-16**). For the HFR diet, 100 g of 60% high-fat diet
in pellets were smashed into powder-like small pieces before 0.5 g of resveratrol was scattered to
the diet by several batches. The mixture was stirred thoroughly from the bottom to the top in a
blender, made into pellets by compression molding and stored at -20°C until use. The diet was
served under a hand-made light-proof shelter and changed twice a week.

*2. In page 5 second paragraph, the full name of TRL needs to be provided earlier.*

**Thanks.** We have revised it accordingly. TRL appears first time in this revised version on **page 3,**
**line 22.**

*3. In Figure 1G (could be 1J), the postprandial TG level in HFR seems significantly lower than*
*that in the LFD control group. This result needs explanation.*

**Thanks** for raising this critique. Yes, we did notice this. Firstly, the difference in the postprandial
TG level in LFD and HFR groups did not reach statistical significance, with the p-value of 0.07.
Secondly, we do not intend to use this single result to claim that REV-I makes the mice even better
at chylomicron secretion. We have revised our description of the result on this aspect (**page 5,**
**lines 17-20**).

Reviewer #2 (Remarks to the Author):

*1. The noteworthy results: the role of jejunum SR-B1-mediated chylomicrons in REV-mediated*
*lipid homeostasis.*

**Thanks** for the encouragement from Expert Reviewer 2.

2. *Some major issues need to be carefully addressed.*

2.1 *The analytical performance of the metabolomic approach.*

*It is recommended that QC samples should be located closely, and that 60% of the metabolites*
*should have a CV value < 15% in the metabolomic data, with a more relaxed criterion (20%) for*
*the metabolites near their limit of quantification.1, 2, 3 It is a low requirement that ‘More than*
*90% of QC samples have a CV value < 0.3’. Therefore, the distribution of QC samples in the score*
*plot of PCA based on the information of all samples (including QC and analytical samples) should*
*be provided to evaluate the analytical performance of the metabolomic approach. In addition,*
*what is the proportion of metabolites with the CV value less than 15%, 20% and 30%, respectively,*
*among all detected metabolites in QC samples?*

*References*

1. Gika HG, Theodoridis GA, Wingate JE, Wilson ID. *Within-day reproducibility of an HPLC-MS-*
*Based method for metabonomic analysis: application to human urine. J Proteome Res* 6, 3291-
3303 (2007).

2. Begley P, et al. *Development and performance of a gas chromatography-time-of-flight mass*
*spectrometry analysis for large-scale nontargeted metabolomic studies of human serum. Anal*
*Chem* 81, 7038-7046 (2009).

3. FDA U. *Guidance for industry: bioanalytical method validation. US Department of health and*
*human services, US FDA. Center for Drug Evaluation and Research, Rockville, MD, USA, (2001).*

**Thanks for raising those important technical issues.** We have studied the above literature
provided by Expert Reviewer 2. In this revised version, we have provided a revised PCA plot with
the distribution of QC samples in Fig. S6A, and it shows that QC samples are located closely. A
total of 2053 metabolites were detected, among which 1723 (83.93%) metabolites have a CV value
< 15%, 1849 (90.06%) have a CV value < 20%, and 1978 (96.35%) have a CV value < 30%.
Therefore, we have revised the description in the methods that more than 80% of the metabolites
in QC samples have a CV value < 15%. Besides, we have cited the above literature 1 and 2 in our
method session.

2.2 *Metabolite identification.*

*It is known that multiple ions would be produced from a metabolite after ionization. The authors*
*pointed out that “Through LC-MS/MS analysis, we detected 2053 metabolites.” In line 253. Is*
*‘2053’ the number of metabolite or that of metabolite ions? If ‘2053’ is the number of metabolites,*
*all the detected metabolites should be listed in the supplementary material, as well as key*
*parameters for the identification, including the retention time, theoretical accurate mass,*
*measured accurate mass, and mass deviation. Besides, metabolites verified by the reference*
*standards should be labeled. Changes in differential metabolites should be provided in the*
*supplementary material, including fold changes, P value, and parameters for the identification*
*mentioned above.*

**We thank Expert Reviewer 2 for the above questions and suggestions.** 2053 is the number of
metabolites the system detected. The theoretical molecular mass, measured precursor ion mass,
ionization model and molecular formula were all listed in
*Data_S2_ALL_sample_data_untargeted.xlsx* in supplemental materials. Specifically, metabolites
verified by the reference standards were labeled as Y in the column with the title name *Standard*.
However, the retention time for metabolite identification is confidential database information of
Metware Biotechnology Co., Ltd, which helped us perform metabolomics profiling and data
analyses. Besides, key parameters of differential metabolites between HFD and HFR or between
HFD and HRH groups have also been listed in *Data_S3_HFD_vs_HFR_difference.xlsx* and
*Data_S4_HFD_vs_HRH_difference.xlsx*, respectively, in supplemental materials.

*2.3 Figure 7.*

*Why fecal extracts from LFD were not analyzed using the metabolomic approach? Since the*
*authors pointed out that ‘Dietary resveratrol intervention improves lipid homeostasis via*
*attenuating HFD-induced fecal chenodeoxycholic acid and jejunum SR-B1 elevation’, the change*
*in CDCA induced by HFD is essential. Therefore, alterations in fecal metabolites in HFD*
*compared to LFD should be provided, where metabolomic data of the LFD group are essential.*
*In addition, changes in fecal bile acids (including CDCA and DCA) from LFD, HFD, HFR to HRH*
*should be clearly presented, which could be provided in the way as Figure 7D, and 7E.*

**We thank Expert Reviewer 2** for raising this important issue. When the metabolomics
experiments were designed, we are somehow curious about the identification of heat stable entities
within the feces that differ between HFD feeding and HFD+REV-I (HFR). For this reason, the
three groups of fecal samples are that of HFD, HFR, and heat treated HFR (HRH). Fortunately,
we identified CDCA and DCA which were higher in the HFD group and lower in both HFR and
HRH groups. We then verified the role of CDCA on FXR and its downstream targets and on SR-
B1. We have conducted further wet-bench work in addressing the above critiques. We have
directly measured BA levels in mouse feces and serum. For this new set of experiment, we
followed the advice by including samples from the LFD group (Fig. 7C-F). Results obtained allow
305 us to claim that HFD feeding increased fecal CDCA and DCA levels, while REV-I attenuated the
306 increases. Since our current study is focussing on fecal CDCA, FXR and SR-B1, it may not be
necessary to conduct further metabolomics analyses, including the LFD group. We ask the
permission to treat this as a future work, in systematic metabolomics profiling with various dietary
challenges and various polyphenol interventions.

We have followed the above advice, presented changes in total fecal bile acids, CDCA and DCA
from LFD, HFD, HFR and HRH groups (Fig. 7C-D). The analyses on other bile acids were
conducted in serum samples in these three groups of mice (Fig. 7E-F).

*2.4 Lines 90-290.*

*The authors tied to demonstrated the involvement of gut microbiota in REV-mediated lipid*
*homeostasis via regulating jejunal SR-B1. Why microbial changes in jejunal contents were not*
*detected? Why metabolomic alterations in jejunal contents were neither conducted? Instead,*
*metabolome in feces and microbiome in cecal/fecal contents were determined. The function of the*

*jejenum identified by the authors does not specifically correspond to changes in fecal metabolome*
*and alterations in microbiome in fecal/cecal contents.*

**Thanks** for raising those important critiques. To take mouse jejunal content for conducting gut
microbiome analyses is technically difficult. We have tried to take and assess those watery
samples. Even though we can collect some, bacterial contents vary substantially among the
samples. We ask the permission for further expanded investigations on jejunal microbial content
changes as a future direction, to be conducted in certain larger animal models, such as rats and
hamsters. We may need to use the cannulation tool to collect those watery content continuously.
We need recruit collaborators for conducting such experiments.

Although the samples we collected were not suitable and enough for our gut microbiome analysis;
and may not be suitable for our metabolomics analysis (we are afraid that rapid turn-over on such
watery samples may cause huge variation among the samples), we have tested the effect of diluted
sterile jejunal extract (the same method of making sterile fecal extract) on SR-B1 expression in
Caco-2 cells. The results showed that jejunal content extract diluted at 1:600 reduced SR-B1
expression. Below is the Western blotting result for Reviewer's eyes only, as include this
premature data is unnecessary, in our view. We will use the above important critiques for
conducting our future work, including the assessment of levels of CDCA and other BAs, as well
as SR-B1 levels at different segment of gut, most likely, in rat or hamster models.

*As a primary bile acid, CDCA is synthesized in the liver and further metabolized into secondary*
*bile acid in the intestine. Therefore, the increase in fecal CDCA in HFD group is related to the*
*increase in the synthesis in the liver and/or the decrease in intestinal microbial degradation. To*
*prove the important role of CDCA elevation mediated by microbial metabolism, it is necessary to*
*find the key microbial enzymes degrading CDCA and verify it through the regulation of related*
*enzymes. There is still much work to be done in this work on the role of microbial metabolism and*
*CDCA in REV-mediated lipid homeostasis via suppressing jejunal SR-B1 expression.*

**We thank Expert Reviewer 2** for the above insightful critique and experimental suggestions. We
have followed the above guidance in conducting our bench work revision. As we have stated in
the beginning of the letter, we have confirmed by conducting bile acid analyses in fecal samples
and in serum, clarified the involvement of both liver production and intestinal microbiome
mediated modification (involving reduced BSH activity and density of BSH-producing bacterial
genus) (**Fig. 7G-I, and Fig. S7**). Meanwhile, we have also treated mice with resveratrol and FXR

agonist GW4064 and found out that the effects of REV-I on chylomicron secretion and gut SR-B1
expression were abrogated by activation of FXR (**Fig. 8F-I**). These results allow us to suggest that
REV-I reduces BSH-producing bacterial genus and BSH activity, leading to less de-conjugation
of TCDCA into CDCA and reduced FXR activity, which inhibits intestinal SR-B1 expression and
chylomicron secretion. We have also followed the advice in measuring liver CADA level and
expression of genes on bile acid synthesis.

**References**

- 1 Huang, F. *et al.* Theabrownin from Pu-erh tea attenuates hypercholesterolemia via modulation
of gut microbiota and bile acid metabolism. *Nat Commun* **10**, 4971, doi:10.1038/s41467-019-
12896-x (2019).
- 2 Chao, F. *et al.* Upregulation of scavenger receptor class B type I expression by activation of FXR
in hepatocyte. *Atherosclerosis* **213**, 443-448, doi:10.1016/j.atherosclerosis.2010.09.016 (2010).
- 3 Tian, L. *et al.* The developmental Wnt signaling pathway effector beta-catenin/TCF mediates
hepatic functions of the sex hormone estradiol in regulating lipid metabolism. *PLoS Biol* **17**,
e3000444, doi:10.1371/journal.pbio.3000444 (2019).
- 4 Bowers, J. L., Tyulmenkov, V. V., Jernigan, S. C. & Klinge, C. M. Resveratrol acts as a mixed
agonist/antagonist for estrogen receptors alpha and beta. *Endocrinology* **141**, 3657-3667,
doi:10.1210/endo.141.10.7721 (2000).
- 5 Dash, S., Xiao, C., Morgantini, C., Szeto, L. & Lewis, G. F. High-Dose Resveratrol Treatment for 2
Weeks Inhibits Intestinal and Hepatic Lipoprotein Production in Overweight/Obese Men.
*Arterioscler Thromb Vasc Biol* **33**, 2895-2901 (2013).
- 6 Jia, W., Xie, G. & Jia, W. Bile acid-microbiota crosstalk in gastrointestinal inflammation and
carcinogenesis. *Nat Rev Gastroenterol Hepatol* **15**, 111-128, doi:10.1038/nrgastro.2017.119
(2018).
- 7 Takahashi, S. *et al.* Cyp2c70 is responsible for the species difference in bile acid metabolism
between mice and humans. *J Lipid Res* **57**, 2130-2137, doi:10.1194/jlr.M071183 (2016).
- 8 Zhang, Y. *et al.* Activation of the nuclear receptor FXR improves hyperglycemia and
hyperlipidemia in diabetic mice. *Proc Natl Acad Sci U S A* **103**, 1006-1011,
doi:10.1073/pnas.0506982103 (2006).
- 9 Zhang, Y. *et al.* Identification of novel pathways that control farnesoid X receptor-mediated
hypocholesterolemia. *J Biol Chem* **285**, 3035-3043, doi:10.1074/jbc.M109.083899 (2010).
- 10 Wang, Y. D. *et al.* Farnesoid X receptor antagonizes nuclear factor kappaB in hepatic
inflammatory response. *Hepatology (Baltimore, Md.)* **48**, 1632-1643, doi:10.1002/hep.22519
(2008).
- 11 Liu, H., Pathak, P., Boehme, S. & Chiang, J. L. Cholesterol 7alpha-hydroxylase protects the liver
from inflammation and fibrosis by maintaining cholesterol homeostasis. *J Lipid Res* **57**, 1831-
1844, doi:10.1194/jlr.M069807 (2016).
- 12 Zhang, Y. *et al.* Bile acids evoke placental inflammation by activating Gpbar1/NF-kappaB
pathway in intrahepatic cholestasis of pregnancy. *J Mol Cell Biol* **8**, 530-541,
doi:10.1093/jmcb/mjw025 (2016).
- 13 Kakiyama, G. *et al.* Modulation of the fecal bile acid profile by gut microbiota in cirrhosis. *J*
*Hepatol* **58**, 949-955, doi:10.1016/j.jhep.2013.01.003 (2013).
- 14 Beaudoin, J. J., Brouwer, K. L. R. & Malinen, M. M. Novel insights into the organic solute
transporter alpha/beta, OSTalpha/beta: From the bench to the bedside. *Pharmacol Ther* **211**,
107542, doi:10.1016/j.pharmthera.2020.107542 (2020).

REVIEWER COMMENTS

Reviewer #1 (Remarks to the Author):

The authors have addressed the previous comments by additional experiments and/or writing revisions. However, there are still some critical questions remaining, which need to be addressed with further clarifications.

Major comments:

1. A potential role of intestine SR-B1 in mediating resveratrol effects on lipid metabolism is nicely demonstrated. In addition, a potential role of gut microbiota and their metabolites in suppressing the expression of intestinal SR-B1 is established. However, the direct link between CDCA-FXR/NF-kB and SR-B1 to explain the mechanism by which REV and microbiota metabolites suppress SR-B1 is not convincing.
2. The authors' explanations that CDCA may activate NF-kB via nuclear receptor/s other than FXR and FXR may be a cofactor of NF-kB have no supporting data and may generate confusions. NF-kB can be activated by many signals instead of CDCA, which come from gut microbiota during HFD feeding. Authors do not need to highlight the effects of CDCA as the only one signal to make a connection to NF-kB and SR-B1. In Fig.8J, NF-kB does not need to be included there.
3. In Fig.7F, 7 major bile acids are shown. However, another key bile acid, LCA and TLCA, is missing.
4. Although CDCA and DCA are heat-stable, there are many other potential metabolites from gut can be heat-stable. In figure 7A and B, there seems many stronger hits other than CDCA and DCA are identified.
5. In the gut, after gut microbiota metabolism, the bile acid species can be expanded to many different types. Recent studies show that they can either activate or suppress several bile acid receptors in addition to FXR, such as TGR5, VDR, PXR, S1PR2. It is reasonable that authors focus on FXR because previous reports indicate FXR activation up regulates SR-B1 in the liver. However, it is not necessary only CDCA can activate FXR in this model. Previous reports have used the extraction of total bile acids from feces and test their effects on FXR activity (using the WT FXRE-Luc and mutant FXRE-Luc reporter assays), which may be more relevant to resveratrol effects.
6. An intestinal FXR tissue-specific mouse line will be the better approach to show the requirement of intestinal FXR in mediating the resveratrol effects in regulating SR-B1 metabolic effects.

Overall, authors can focus more on the roles of intestinal FXR in mediating the resveratrol effects on SR-B1 expression and lipid metabolism instead of focusing on CDCA. Also, intestinal NF-kB lacks supporting data in this paper and can be used as a future direction.

Reviewer #2 (Remarks to the Author):

Regarding the retention time of metabolites. It is strongly recommended to provide the retention time of metabolites, which is an important parameter for metabolite identification. For example, polar metabolites are eluted before non-polar metabolites after the separation by a reversed-phase column. The retention time can be used to exclude some unlikely metabolites in the sample. In addition, more than 2000 metabolites were identified in this study. The number of identified metabolites is larger than that in many studies, so the identification result will provide helpful references for many researchers, and increase the impact of this work.

Reviewer #1 (Remarks to the Author):

The authors have addressed the previous comments by additional experiments and/or writing revisions. However, there are still some critical questions remaining, which need to be addressed with further clarifications.

Major comments:

1. A potential role of intestine SR-B1 in mediating resveratrol effects on lipid metabolism is nicely demonstrated. In addition, a potential role of gut microbiota and their metabolites in suppressing the expression of intestinal SR-B1 is established. However, the direct link between CDCA-FXR/NF- κ B and SR-B1 to explain the mechanism by which REV and microbiota metabolites suppress SR-B1 is not convincing.

We thank Reviewer 1 for the nice summary and encouragement. We totally agree with the above critique and have made further revision in our manuscript, stating that detailed relationship among FXR, NF- κ B, CDCA and SR-B1 needs intensive further investigations with global effort (page 13, lines 14-16). We have also followed the advice, not over-emphasizing CDCA in the entire manuscript (including its title) as the direct role of CDCA needs further *in vivo* investigations. We have now provided detailed rationale that why CDCA was picked as an example bile acid for our current investigation (page 13, line 2; page 14, lines 18-21). Finally, we have also stated that CDCA or even bile acids are not sole players that mediate function of REV-1 on repressing intestinal SR-B1 in the discussion, other differentially expressed metabolites revealed by our metabolomics analysed should also be investigated in the future (page 18, lines 16-18).

2. The authors' explanations that CDCA may activate NF- κ B via nuclear receptor/s other than FXR and FXR may be a cofactor of NF- κ B have no supporting data and may generate confusions. NF- κ B can be activated by many signals instead of CDCA, which come from gut microbiota during HFD feeding. Authors do not need to highlight the effects of CDCA as the only one signal to make a connection to NF- κ B and SR-B1. In Fig.8J, NF- κ B does not need to be included there.

Thanks for the above critique. In response to the critique by the Expert Reviewer in the 1st round regarding relationships among FXR, NF- κ B and SR-B1, within the 3-month allowed period, we have conducted our explorations utilizing chemical inhibitors for NF- κ B or FXR in the Caco-2 cell line without and with CDCA treatment. Data obtained indicated the existence of complicated relationship among them. In our view, we have provided our interpretations of those bench work results clearly and meaningfully, with proper literature citation and discussion (page 13, lines 1-16; Ref. 29-33). We agree that we cannot make a final conclusion on those observations. We sincerely ask the permission to keep those data (Figure 8B/C/D) and our explanations within the manuscript, as some of our peers may be interested in those findings. We have followed the advice of Expert Reviewer 1 to remove NF- κ B in Figure 8J.

3. In Fig.7F, 7 major bile acids are shown. However, another key bile acid, LCA and TLCA, is missing.

Thanks for the above critique. We have conducted our measurement on serum LCA and TLCA. The changes in LCA level by HFD or HFR treatment were similar as that of CDCA,

while there was no significant change in serum TLCA levels among the three groups. We hence have revised Figure 7F with the addition of the new data.

4. Although CDCA and DCA are heat-stable, there are many other potential metabolites from gut can be heat-stable. In figure 7A and B, there seems many stronger hits other than CDCA and DCA are identified.

Thanks for the above critique. Those hits are mainly derivatives of amino acids with no known biological functions yet. We totally agree that the functions of those hits should be assessed in our future studies (page 19, lines 5-11). A list of those metabolites are shown in the supporting files (Data_S2_ALL_sample_data_untargeted.xlsx).

5. In the gut, after gut microbiota metabolism, the bile acid species can be expanded to many different types. Recent studies show that they can either activate or suppress several bile acid receptors in additional to FXR, such as TGR5, VDR, PXR, S1PR2. It is reasonable that authors focus on FXR because previous reports indicate FXR activation up regulates SR-B1 in the liver. However, it is not necessary only CDCA can activate FXR in this model. Previous reports have used the extraction of total bile acids from feces and test their effects on FXR activity (using the WT FXRE-Luc and mutant FXRE-Luc reporter assays), which may be more relevant to resveratrol effects.

We highly appreciate the above comments and advice, which will help us to conduct future investigations, including detailed role of each bile acid (and other heat stable metabolites) in the feces that are modified by HFD feeding and REV-I, utilizing tools including Luc reporter assay (for each of the above nuclear receptors).

6. An intestinal FXR tissue-specific mouse line will be the better approach to show the requirement of intestinal FXR in mediating the resveratrol effects in regulating SR-B1 metabolic effects.

We thank Expert Reviewer 1 for the above advice. We did think about this during our last round revision process. However, to obtain such mouse line, to establish the methodology in handling such mouse line, and to determine metabolic profile of this mouse line in the absence and presence of HFD challenge, without and with REV-I will take a lot of time. In addition, we don't know whether such knockout mouse line shows different gut microbiome profiles, which adds further complexity in our experimental design. Finally, FXR is a multiple functional nuclear receptor. Its absence in the gut may affect other signaling cascades. We hence conducted its functional activation by gavage, which blocked the beneficial effect of REV-I on attenuating gut SR-B1. We will think about how FXR tissue-specific mouse lines can be utilized in our future investigations.

Overall, authors can focus more on the roles of intestinal FXR in mediating the resveratrol effects on SR-B1 expression and lipid metabolism instead of focusing on CDCA. Also, intestinal NF-kB lacks supporting data in this paper and can be used as a future direction.

We thank Expert Reviewer for the above advice 1 and have conducted our editorial revision accordingly. As stated above, we have removed NF-kB in Figure 8J. We ask the permission to

keep the data in Figure 8B/C/D within the manuscript. This is also because in this manuscript we have provided ChIP and qChIP observation with mouse gut tissue samples that NF-kB binds to SR-B1 promotor and activates SR-B1 transcription (entire Figure 5). We agreed that complicated relationships among FXR, NF-kB and SR-B1 should be assessed in future studies.

Reviewer #2 (Remarks to the Author):

Regarding the retention time of metabolites. It is strongly recommended to provide the retention time of metabolites, which is an important parameter for metabolite identification. For example, polar metabolites are eluted before non-polar metabolites after the separation by a reversed-phase column. The retention time can be used to exclude some unlikely metabolites in the sample. In addition, more than 2000 metabolites were identified in this study. The number of identified metabolites is larger than that in many studies, so the identification result will provide helpful references for many researchers, and increase the impact of this work.

We thank Expert Reviewer 2 for the above advice. As we have indicated in the method session, the metabolomics assessment was conducted by Wuhan Metware Biotechnology Co., Ltd. (Wuhan, China). The metabolites were identified by searching for the internal database and public databases (MassBank, KNApSAcK, HMDB, MoTo DB, and METLIN). For the internal database, it was constructed based on the standard materials and purified compounds. Additionally, certain public databases (MassBank, KNApSAcK, HMDB, MoTo DB, and METLIN) also contain relevant information of metabolites that can be referenced directly. The metabolites were identified by comparing the accurate precursor ion (Q1) and production (Q3) values, retention time, and fragmentation pattern with the database. Many research groups have cooperated with Wuhan Metware Biotechnology Co., Ltd. and utilized this database to identify metabolites. Below are two example recent publications that utilized the service from this company: 1) Gut, 2021, 70:2297-2306; and 2) Signal Transduct Target Ther, 2021, 6:345.

In terms of the retention time, we have identified metabolites by comparing the ion pairs, retention time, and fragmentation pattern in the database. We totally agree with Expert Reviewer 2 that retention time is very important for the annotation of each metabolite. Unfortunately, due to the commercial conflict of interest, our cooperater cannot provide the entire information about their retention time. The company agreed to provide us the retention time of 30 metabolites which are considered as the key differential metabolites with high VIP values. We have listed the information in Data_S5_Retention_Time.xlsx in supplemental tables.

REVIEWERS' COMMENTS

Reviewer #1 (Remarks to the Author):

The authors have addressed most of the previous comments. Overall, this is a nice work to link gut microbiota-bile acids to the biological effects of a well-known small molecule compound.

Reviewer #2 (Remarks to the Author):

Regarding the retention time of identified metabolites. To my knowledge, the retention times of metabolites could be provided by many companies serving metabolomics analysis. The retention time of metabolite provided by many companies is no secret. Without the retention time, the metabolite identification table in this study will not be of much reference significance for the study of others.

REVIEWERS' COMMENTS

Reviewer #1 (Remarks to the Author):

The authors have addressed most of the previous comments. Overall, this is a nice work to link gut microbiota-bile acids to the biological effects of a well-known small molecule compound.

We thank the Expert Reviewer 1 for the nice comments and encouragements.

Reviewer #2 (Remarks to the Author):

Regarding the retention time of identified metabolites. To my knowledge, the retention times of metabolites could be provided by many companies serving metabolomics analysis. The retention time of metabolite provided by many companies is no secret. Without the retention time, the metabolite identification table in this study will not be of much reference significance for the study of others.

We thank the Expert Reviewer 2 for the advice and agree with his/her view that “the retention times of metabolites could be provided by the service company”. As we have explained previously, we have made our effort in persuading the company to provide such information for our readers. To this end, we have made the effort in providing retention times for 30 key differential metabolites (as shown in *Data_S5_Rentition_Time.xlsx*) discussed in our current study, and in providing detailed information about the service company (including its address and web-site information, etc.). We have also provided the information that this company has assisted other research teams in conducting similar metabolomics analyses without providing the retention time for identified metabolites (Gut. 2021, 70: 2297-2306; Signal Transduct Target Ther. 2021, 6: 345.). In our future study, we will continue work on other metabolites that have been identified. Hopefully we can persuade this company to change their current policy.